# Presynaptic quantal size enhancement counteracts post-tetanic release depression

Anu G. Nair[1,2] , Nasrin Bollmohr[1,4,5] , Levin Schökle[1] , Jennifer Keim[1], José María Mateos Melero[3] and Martin Müller[1,4,5]

[1]*Department of Molecular Life Sciences, University of Zurich, Zurich, Switzerland*
[2]*Department of Neuroscience, Karolinska Institutet, Stockholm, Sweden*
[3]*Center for Microscopy and Image Analysis, University of Zurich, Zurich, Switzerland*
[4]*Neuroscience Center Zurich, University of Zurich/ETH Zurich, Zurich, Switzerland*
[5]*University Research Priority Program (URPP)Adaptive Brain Circuits in Development and Learning (AdaBD)University of Zurich, Zurich, Switzerland*

Handling Editors: Katalin Toth & Samuel Young

The peer review history is available in the Supporting Information section of this article (https://doi.org/10.1113/JP286176#support-information-section).

**Abstract** Repetitive synaptic stimulation can induce different forms of synaptic plasticity but may also limit the robustness of synaptic transmission by exhausting key resources. Little is known about how synaptic transmission is stabilized after high-frequency stimulation. In the present study, we observed that tetanic stimulation of the *Drosophila* neuromuscular junction (NMJ) decreases quantal content, release-ready vesicle pool size and synaptic vesicle density for minutes after stimulation. This was accompanied by a pronounced increase in quantal size. Interestingly, action potential-evoked synaptic transmission remained largely unchanged. EPSC amplitude fluctuation analysis confirmed the post-tetanic increase in quantal size and the decrease in quantal content, suggesting that the quantal size increase counteracts release depression to maintain evoked transmission. The magnitude of the post-tetanic quantal size increase and release depression correlated with stimulation frequency and duration, indicating activity-dependent stabilization of synaptic transmission. The post-tetanic quantal size increase persisted after genetic ablation of the

Anu G. Nair obtained his PhD from the KTH Royal Institute of Technology, Stockholm, and conducted his postdoctoral research on homeostatic regulation of synaptic transmission at the University of Zurich, Switzerland. Currently, he is a scientist in the preclinical phase of drug development in the pharmaceutical industry, where he employs mathematical modelling to enable clinical translation. Nasrin Bollmohr completed her MSc in Biochemistry at Freie Universität Berlin. During her master's thesis at the Bordeaux Neurocampus, she studied lactate-dependent coupling of energy metabolism and gliotransmission. Intrigued by synaptic processes, she joined Martin Müller's lab at the University of Zurich for her PhD to explore how synaptic activity shapes synaptic nano-architecture and function.

A. G. Nair and N. Bollmohr contributed equally to this work.

The Journal of Physiology

glutamate receptor subunits GluRIIA or GluRIIB, and glutamate receptor calcium permeability, as well as blockade of postsynaptic calcium channels. By contrast, it was strongly attenuated by pharmacological or presynaptic genetic perturbation of the GTPase dynamin. Similar observations were made after inhibition of the $H^+$-ATPase, suggesting that the quantal size increase is presynaptically driven. Additionally, dynamin and $H^+$-ATPase perturbation resulted in a post-tetanic decrease in evoked amplitudes. Finally, we observed an increase in synaptic vesicle diameter after tetanic stimulation. Thus, a presynaptically-driven quantal size increase, likely mediated by larger synaptic vesicles, counterbalances post-tetanic release depression, thereby conferring robustness to synaptic transmission on the minute time scale.

(Received 25 April 2024; accepted after revision 30 July 2024; first published online 24 August 2024)

**Corresponding author** M. Müller: Department of Molecular Life Sciences, University of Zurich, Winterthurerstrasse 190, 8057 Zurich, Switzerland.    Email: martin.mueller@mls.uzh.ch

**Abstract figure legend** Sustained high-frequency stimulation decreases the number of synaptic vesicles that fuse per action potential at the *Drosophila* neuromuscular junction. However, the postsynaptic response remains largely unchanged. An increased postsynaptic response to individual vesicles, likely driven by an increase in vesicle size resulting from endocytosis defects, stabilizes synaptic efficacy for minutes after sustained activity. Our findings advance our understanding of the mechanisms promoting robust synaptic transmission during sustained neuronal activity.

## Key points

- Many synapses transmit robustly after sustained activity despite the limitation of key resources, such as release-ready synaptic vesicles.
- We report robust synaptic transmission after sustained high-frequency stimulation of the *Drosophila* neuromuscular junction despite a reduction in release-ready vesicle number.
- An increased postsynaptic response to individual vesicles, likely driven by an increase in vesicle size due to endocytosis defects, stabilizes synaptic efficacy for minutes after sustained activity.
- Our study provides novel insights into the mechanisms governing synaptic stability after sustained neural activity.

## Introduction

Although repetitive synaptic stimulation can induce different forms of synaptic plasticity, it may also limit the robustness of synaptic transmission by exhausting essential resources. Nonetheless, many synapses transmit robustly at action potential (AP) rates of tens to hundreds of Hertz (Kim et al., 2013; Lorteije et al., 2009; Ormerod et al., 2022; Ritzau-Jost et al., 2014). A major constraint that limits synaptic transmission during and after sustained activity is the availability of release-ready synaptic vesicles (Betz, 1970; Rosenmund & Stevens, 1996; Storozhuk et al., 2002; Thomson et al., 1993; von Gersdorff et al., 1997). Several mechanisms have been proposed to counteract the depletion of release-competent synaptic vesicles on different time scales. On the millisecond to second time scale, the synaptic vesicle priming factor UNC13-1, the presynaptic scaffold Bassoon or the $Ca^{2+}$-sensor synaptotagmin-3 have been implicated in promoting the resupply of synaptic vesicles at various synapses (Chen et al., 2013; Frank et al., 2010; Weingarten et al., 2022). Further mechanisms that have been suggested to counteract release depression within milliseconds to seconds include kiss-and-run exocytosis (Ceccarelli et al., 1972; de Toledo et al., 1993; Fesce et al., 1994; Stevens & Williams, 2000), and ultrafast endocytosis (Watanabe et al., 2013). On the time scale of seconds to minutes, clathrin-mediated endocytosis (Dittman & Ryan, 2009; Heuser & Reese 1973; Pearse, 1976), bulk endocytosis (Cheung et al., 2010; Clayton et al., 2007) and mobilization of synaptic vesicles from a reserve pool of synaptic vesicles (Humeau et al., 2001; Kuromi & Kidokoro, 1998; Prado et al., 1992) were shown to stabilize synaptic transmission. Although these mechanisms potently counteract release depression, sustained synaptic stimulation at high rates eventually saturates a synapse's capacity to replenish release-ready synaptic vesicles and causes depression of evoked postsynaptic responses. Little is known about the mechanisms that stabilize synaptic efficacy when the machinery supporting the resupply of release-ready vesicles is saturated.

High-frequency or "tetanic" stimulation induces post-tetanic potentiation (PTP), a transient increase in evoked amplitudes on a time scale of seconds to minutes, at various synapses (Delaney et al., 1989; Korogod et al., 2005; Zucker & Regehr, 2002). There is also some evidence that tetanic stimulation can result in post-tetanic depression of AP-evoked transmission (Betz, 1970; Guo & Zhong, 2006; Neale et al., 2001; Storozhuk et al., 2002; von Gersdorff et al., 1997). At the level of miniature synaptic transmission, several studies reported a post-tetanic increase in miniature frequency and amplitude on the second to minute time scale (Jorquera et al., 2012; Kombian et al., 2000; Korogod et al., 2005; Quinlan & Hirasawa, 2013). The increased miniature frequency and amplitude during PTP are considered to reflect enhanced release probability (Jensen et al., 1999; Mahapatra & Lou, 2017), compound fusion of synaptic vesicles (He et al., 2009), and multivesicular release (Kombian et al., 2000; Quinlan & Hirasawa, 2013). At the *Drosophila* neuromuscular junction (NMJ), protein kinase A-dependent phosphorylation of complexin promotes post-tetanic potentiation of miniature frequency (Cho et al., 2015). The post-tetanic increase in release probability and quantal size inferred from miniature synaptic transmission may contribute to robust evoked transmission during sustained synaptic activity. However, the relationship between miniature synaptic transmission and the maintenance of evoked responses after repetitive synaptic activity has remained largely unexplored.

In the present study, we investigated synaptic adaptations in response to sustained synaptic activity and how those adaptations are linked to the post-tetanic robustness of synaptic transmission at the *Drosophila* NMJ using electrophysiology, confocal microscopy, and transmission electron microscopy. Despite transient release depression and synaptic vesicle depletion following tetanic stimulation, we observed sustained maintenance of evoked responses. Tetanic stimulation increased quantal size for minutes after stimulation. Moreover, we observed larger vesicles after the tetanus. Interfering with synaptic vesicle recycling and refilling attenuated the post-tetanic quantal size increase and caused depression of evoked amplitudes. Together, we propose that a post-tetanic increase in quantal size, likely driven by larger synaptic vesicles due to endocytosis defects, stabilizes synaptic transmission after sustained activity on the minute time scale.

## Methods

### *Drosophila* stocks

Experiments involving genetically modified *Drosophila* were approved by the responsible authorities (Authorization A120910-09). *Drosophila melanogaster*

stocks were raised on standard molasses-containing food at 25°C and constant humidity under a 12:12 h light/dark photocycle. Third-instar larvae of the following lines were used (Bloomington Drosophila Stock Centre, BDSC; FlyBase, FB): *w1118* (BDSC 3605, FBal0018186), *GluRIIA^SP16* (Graeme Davis lab, FBal0085982), *GluRIIB^SP5* (Dion Dickman lab; Muttathukunnel et al., 2022), *GluRIIA^Q615R* and *GluRIIB^SP6* (Han et al., 2023), and *UAS-shibire^ts* (Jan Pielage lab; Ramaswami et al., 1993). Flies of the *OK371-Gal4* driver line (BDSC 26 160) were crossed with *UAS-shibire^ts* flies to overexpress the temperature-sensitive form of dynamin in motoneurons. The rearing temperature of these crosses was set at 18°C to avoid premature impairment of dynamin function. *w1118* is the default genotype used for experiments unless stated otherwise.

### Larval sample preparation

Wandering third-instar larvae were dissected in HL3 solution (5 mm KCl, 70 mm NaCl, 10 mm Na-HEPES, 5 mm HEPES, 5 mm trehalose, 115 mm sucrose and 10 mm $MgCl_2$) with 0.3 mm $CaCl_2$ unless stated otherwise. The internal organs, including the brain and the ventral nerve cord, were carefully removed from the body wall, preserving muscle fibres and innervating motor nerves.

### Electrophysiology

Sharp-electrode recordings were performed on muscle 6 of segments 3 and 4 with sharp borosilicate glass electrodes filled with 3 m potassium chloride (resistance 10–25 MΩ). Signal detection was achieved with a HS 9A ×0.1 headstage, connected to an Axoclamp 900A amplifier (Molecular Devices, San Jose, CA, USA) and signal digitization was performed using an Axon Digidata 1550A (Molecular Devices) at a sampling frequency of 10 kHz through Clampex, version 10.7 (Molecular Devices). EPSPs were induced at individual NMJs by stimulating the respective hemi-segmental nerve with single action potentials using a pulse stimulator (model 2100; A-M Systems, Sequim, WA, USA). The voltage clamp measurements were performed in two-electrode voltage clamp (TEVC) configuration. For TEVC recordings, the HS 9A ×0.1 headstage was combined with a HS 9A ×10 headstage.

For time-course experiments, baseline miniature EPSPs (mEPSPs) and EPSPs (0.2 Hz, 3 ms stimulus duration) were recorded (0.3 mm extracellular $Ca^{2+}$, $[Ca^{2+}]_e$) simultaneously for 5 min before applying the tetanic stimulus, followed by further simultaneous measurements of mEPSPs and EPSPs. For each cell, median mEPSP and EPSP amplitudes were binned at intervals of 35 s. Quantal content was calculated as the ratio between

the median EPSP amplitude and the median mEPSP amplitude of the respective bins. Single and intergroup comparisons were performed using normalized quantities (i.e. the fractional changes in mEPSP, EPSP and QC relative to the pre-tetanus baseline average). The binned time-series for these quantities for each recorded cell were parameterized using a double exponential fit, and the post-tetanic changes were estimated from the fit.

The evoked amplitude fluctuation analysis was done using TEVC with a muscle holding potential of $-80$ mV. 50 EPSCs (0.3 Hz) recorded under low release ($p_r$) conditions (0.25 mM $[Ca^{2+}]_e$, 0.1 ms stimulus duration) were analyzed before and 3 min after the tetanic stimulation. Quantal content was estimated as the inverse of squared coefficient of variance ($1/CV^2 = \mu^2/\sigma^2$), and quantal size, in this case, was estimated as the ratio between EPSC variance and mean ($q = \sigma^2/\mu$) (McLachlan, 1978).

Cumulative EPSC amplitude analysis was done using TEVC with a muscle holding potential of $-60$ mV (2 mM $[Ca^{2+}]_e$) according to Schneggenburger et al. (1999). In brief, EPSCs were recorded during 60 Hz trains for 1 s, and the last 10 cumulative EPSC amplitudes of the train were fitted to a line as a function of time, the back-extrapolated intercept to the EPSC amplitude axis is estimated as the cumulative EPSC amplitude. EPSC amplitudes during 60 Hz stimulation were quantified as the difference between the minimum EPSC amplitude and the baseline before the onset of the respective EPSC.

The following pharmacological reagents were diluted in HL3, and dimethyl sulphoxide (DMSO) if required, and directly applied to the dissected larval preparations before electrophysiological recordings, and were present throughout the entire experiment: Dynasore (50 μM; D7693; Sigma-Aldrich, St Louis, MO, USA), bafilomycin 1A (500 nM; #1334; Tocris, Bristol, UK), nifedipine (25 μM; #1075; Tocris), ryanodine (6.25 μM; #1329; Tocris) and CdCl$_2$ (10 μM; #202908; Sigma-Aldrich). Controls were either incubated with HL3, or HL3 containing the respective DMSO concentration.

For the iontophoretic glutamate puff experiments, a patch pipette filled with HL3 containing glutamate sodium salt (100 μM; #1064451000; Sigma-Aldrich) was positioned near the NMJ between the muscles 6 and 7. A positive current was applied to minimize baseline glutamate leak, and glutamate was then rapidly released by brief negative current pulses (Jan & Jan, 1976). The strength and duration of the negative current pulse were chosen to obtain an PSP amplitude of $\sim$20–25 mV for a single puff. For tetanic stimulation, negative current pulses were applied at 80 Hz for 30 s in the presence of CdCl$_2$ (10 μM) to avoid direct stimulation of presynaptic calcium channels and neurotransmitter release.

For the *shibire$^{ts}$* time-course experiments, a heated water circulation system was installed under the recording chamber to control the bath temperature. The temperature of the HL3 bath was increased to 30°C before the recording at dynamin-expressing or control NMJs. The temperature was maintained and controlled throughout the experiment.

Scientific Python libraries, including numpy, scipy, IPython and neo (Garcia et al., 2014), were used to analyze electrophysiology data. mEPSPs were detected and quantified using a template-matching algorithm (Clements & Bekkers, 1997).

## Electron microscopy

After larval dissection (1 mM $[Ca^{2+}]_e$), the motoneurons of segments 3–5 of one larval side were stimulated at 80 Hz for 30 s, whereas the NMJs on the other side served as unstimulated controls. Larvae were fixed 3 min after stimulation with 2% formaldehyde and 2.5% glutaraldehyde (in 0.2 M phosphate buffer) for 3 h at room temperature. Subsequently, the sample was rinsed with 0.1 M cacodylate buffer and further incubated in 1% osmium tetroxide solution (in 0.1 M cacodylate buffer) for 1 h. After rinsing with ddH$_2$O, contrast enhancement was achieved by immersing the sample in 1% uranyl acetate for 1 h at room temperature, before being stored in ddH$_2$O at 4°C overnight. On the subsequent day, the specimen underwent a dehydration process. This involved the sequential application of ethanol solutions (70% for 20 min, 80% for 20 min, 96% for 25 min and 100% for 25 min), followed by immersion in 100% anhydrous ethanol (2 × 15 min) and immersion in 100% propylene oxide (2 × 15 min). The samples were then placed in a 66% mixture of epon/araldite (in propylene oxide) at room temperature overnight. The following day, samples were treated with a 100% mixture of epon/araldite for 30 min. Finally, samples were embedded in the same solution and incubated in an oven at 60°C overnight to facilitate the polymerization of epoxy resins. Samples were bisected to separate the stimulated and unstimulated sides, then selectively trimmed to retain only muscles 6 and 7 of abdominal segments 3–5. Ultrathin sections of 65 nm thickness were generated using an ultramicrotome (ARTOS 3D; Leica Microsystems, Wetzlar, Germany) and collected onto formvar-coated, carbon-sputtered electron microscopy grids. All sections underwent contrast enhancement via incubation in 1% lead citrate (in H$_2$O) for 7 min, followed by three brief rinses with ddH$_2$O. Images were acquired at 120 kV acceleration voltage in a Talos L120C transmission electron microscope (Thermo Fisher Scientific, Waltham, MA, USA), equipped with a bottom-mounted Ceta camera. Maps software (Thermo Fisher Scientific) was used for automatic acquisition of images with a pixel size of 1.7 nm.

To score synaptic vesicle density at active zones, vesicles were manually counted within a region of interest (ROI) defined by a 300 nm radius around the T-bar centre. Synaptic vesicle densities are probably underestimated because segmentation was confined to clearly discernable vesicles to reduce false positive segmentation errors. Synaptic vesicle outer diameters were quantified by carefully drawing a mask around the outer circumference of each discernible synaptic vesicle in the same ROI around the T-bar centre, or in a 400 nm × 400 nm ROI containing synaptic vesicles placed within the bouton far away from the T-bar. The resulting synaptic vesicle area was then used for the diameter estimate. Large endosome-like structures (vesicle-like structures ≥70 nm) were excluded from the analysis. Segmentation and analyses were conducted manually in a blinded manner by two of the investigators.

### Immunohistochemistry and confocal microscopy

After larval dissections, the motoneurons of segments 3–5 of one larval side were stimulated at 80 Hz for 30 s, whereas the NMJs on the other side served as unstimulated controls. Larvae were fixed 3 min after stimulation for 2 min in Bouin's solution (Sigma-Aldrich) and rinsed three times with phosphate-buffered saline (PBS). Subsequently, larvae were washed thoroughly with PBS containing 0.1% Triton X-100 (PBST) on a rotating wheel. After washing, preparations were blocked with 3% normal goat serum in PBST for 1 h. Larvae were then incubated with primary antibodies (in blocking solution) at 4°C overnight. The following primary antibodies were used: mouse-anti-GluRIIA (dilution 1:100; Developmental Studies Hybridoma Bank, Iowa, IA, USA; 8B4D2 MH2B), rabbit-anti-GluRIIC (dilution 1:100; Sigrist lab, Freie Universität Berlin, Berlin, Germany). The next day, secondary antibodies, anti-mouse-AF555 (dilution 1:500; Life Technologies, Carlsbad, CA, USA; A21424) and anti-rabbit-Abberior Star 635P (dilution 1:200; Abberior Instruments, Göttingen, Germany; 00224JR5) were applied in blocking solution for 1.5 h. The neuronal membrane was stained with anti-Hrp-647 (dilution 1:200; Jackson ImmunoResearch, West Grove, PA, USA; AB_2 338 967). Secondary antibodies were washed off (PBST 5 × 10 min) before preparations were mounted in ProLong Gold Antifade (Thermo Fisher Scientific).

Images were acquired using an inverted Leica SP8 confocal laser scanning microscope (University of Zurich Centre for Microscopy and Image Analysis) controlled with the LASX software suite (Leica Microsystems). Images were acquired using a 100× oil immersion objective (HC PL APO 1.40 NA Oil STED WHITE; Leica Microsystems). Emitted fluorescence was detected with two HyD detectors in photon counting mode (Leica Microsystems).

Microscopy images were analyzed using custom routines written in Fiji, version 1.51n (National Institutes of Health, Bethesda, MD, USA). To quantify GluRIIA and GluRIIC fluorescence intensity, maximum intensity projection images were segmented by binary threshold masks (15% or 35% of the maximum intensity value), background corrected (rolling ball, radius = 1 μm) and filtered (3 × 3 median). Average intensity values were then quantified for each segmented object in background-corrected, unfiltered maximum intensity projection images.

### Statistical analysis

Statistical tests were performed using the stats module of scipy and scikit-posthocs. Two-sided $t$ tests were used to perform two-group statistical comparisons. For assessing changes in the time course data, single sample two-sided $t$ tests were used with the baseline normalized to 1 as the expected population mean for the null hypothesis. The estimated maximum post-tetanic changes normalized to baseline were then tested against the null hypothesis. The distribution of synaptic vesicle diameters was compared between stimulated and unstimulated NMJs using a Kolmogorov–Smirnov test. The exact $P$ values are provided where appropriate. Effect sizes ($d$) were quantified as Cohen's $d$ and are listed where appropriate when a statistically significant difference was observed. The number of samples ($n$) is provided in respective figure captions and text, where $n$ refers to the number of NMJs.

## Results

### A post-tetanic quantal size increase counteracts release depression

To investigate whether synaptic efficacy is stabilized after sustained presynaptic stimulation, we monitored synaptic transmission after tetanic stimulation of the *Drosophila* NMJ. Tetanic stimulation (80 Hz for 30 s) (Fig. 1*A*) increased the amplitude of spontaneous miniature excitatory postsynaptic potentials (mEPSPs) by ∼75% compared to pre-tetanic control levels ($P < 0.001$; $n = 9$) and mEPSP amplitudes returned to baseline within ∼10 min (Fig. 1*B–D*). Consistent with previous reports (Kombian et al., 2000; Quinlan & Hirasawa, 2013), we also observed a significant post-tetanic increase in the mEPSP frequency (see below, Fig. 6*B* and *D*). By contrast, AP-evoked EPSP amplitudes decreased by ∼15% after tetanic stimulation ($P = 0.0146$) (Fig. 1*B–D*). Together, the pronounced post-tetanic increase in mEPSP

amplitude and the relatively small EPSP amplitude decrease translate into a reduction in quantal content by ∼50% (quantal content, QC = EPSP/mEPSP; $P <$ 0.001) (Fig. 1*B–D*), indicating a decrease in presynaptic release. Synaptic transmission remained rather stable during recordings without tetanic stimulation (mEPSP amplitude: $P = 0.127$; quantal content: $P = 0.104$; $n = 6$) (Fig. 1*E*). Collectively, these observations suggest that high-frequency synaptic activity induces a pronounced post-tetanic release depression that is accompanied by an increase in quantal size, whereas AP-evoked synaptic transmission remains largely unchanged. These findings support the idea that the post-tetanic increase in quantal size probably counteracts the release depression, thereby maintaining EPSP amplitude around baseline levels.

## Post-tetanic quantal parameter modulation is activity-dependent

Next, we probed the activity dependence of the post-tetanic modulation of synaptic transmission. First, we investigated synaptic transmission after varying the frequency and the duration of tetanic stimulation (Fig. 2*A*). The magnitude of the post-tetanic increase in mEPSP amplitude and the decrease in quantal content correlated with stimulation frequency ($n = 5, 8, 9$ and $8$) (Fig. 2*B*) and duration ($n = 6, 6, 8$) (Fig. 2*C*), suggesting an activity-dependent modulation of quantal parameters. Second, we tested whether similar modulations can be seen after burst-like stimulation, which resembles the physiological activity patterns of motoneurons and peripheral central pattern generators more closely (Fox et al., 2006; Marder et al., 2022; Newman et al., 2017; Ormerod et al., 2022). Similar to our observations for continuous tetanic, burst-like stimulation (6 × 40 Hz for 5 s, every 5 s) also resulted in a prominent increase in mEPSP amplitude ($P < 0.001$; $n = 7$), whereas AP-evoked EPSP amplitudes remained largely stable ($P = 0.606$), resulting in a decrease in quantal content ($P < 0.001$) (Fig. 2*D–F*). Hence, post-tetanic quantal parameter modulation is activity-dependent, and can be induced by continuous and burst-like tetanic stimulation.

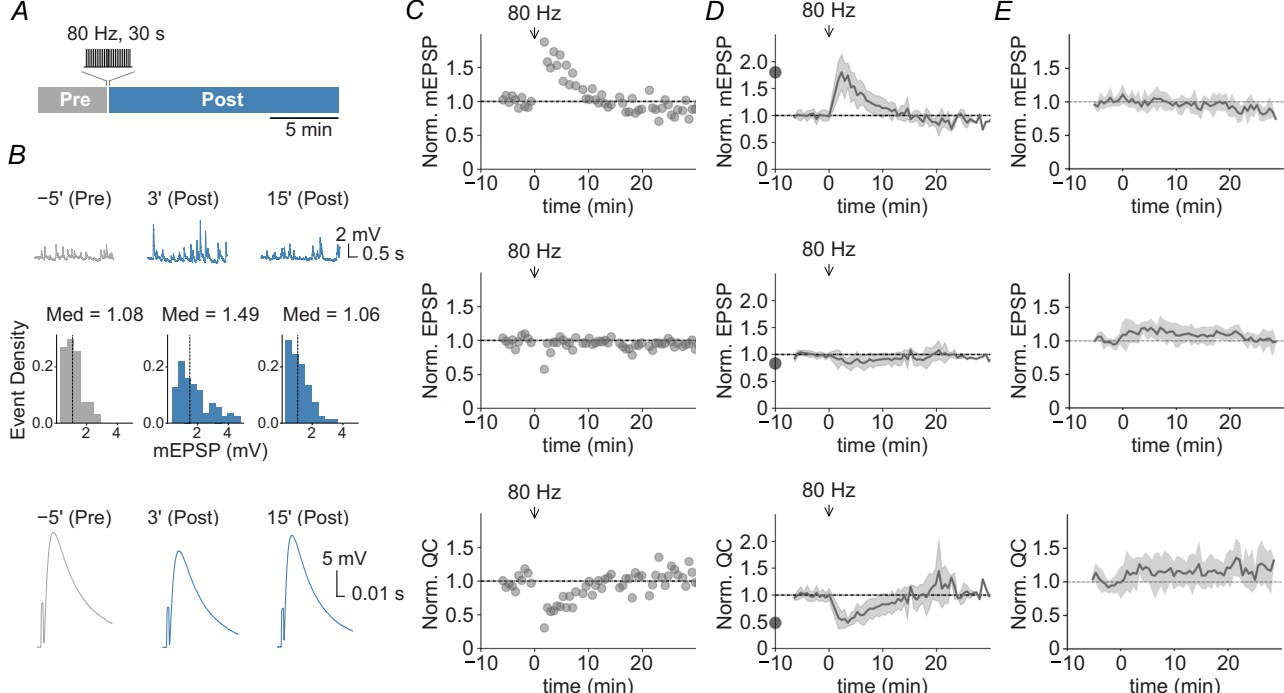

**Figure 1. Increased mEPSP amplitude but unchanged EPSP amplitude after tetanic stimulation**
*A*, experimental design: mEPSPs and EPSPs (0.2 Hz) were recorded before (Pre) and after (Post) tetanic stimulation (80 Hz for 30 s). *B*, representative mEPSPs (top row), mEPSP amplitude densities (along with median value, Med; second row) and EPSP traces (third row) 5 min before (−5′), 3 min after (3′) and 15 min after (15′) tetanic stimulation for a representative cell. *C*, mEPSP amplitude, EPSP amplitude and quantal content (QC = EPSP/mEPSP) values averaged for 35 s bins normalized to the mean pre-tetanic levels as a function of time relative to the onset of tetanic stimulation for the same NMJ. Arrows indicate the timing of tetanic stimulation (80 Hz, 30 s). *D*, corresponding mean ± SD mEPSP amplitude, EPSP amplitude and quantal content values for all NMJs ($n = 9$). The dots on the *y*-axis correspond to the maximal/minimal mean post-tetanic change. *E*, mean ± SD mEPSP amplitude, EPSP amplitude and quantal content values normalized to pre-tetanic mean values for control NMJs without stimulation ($n = 6$).

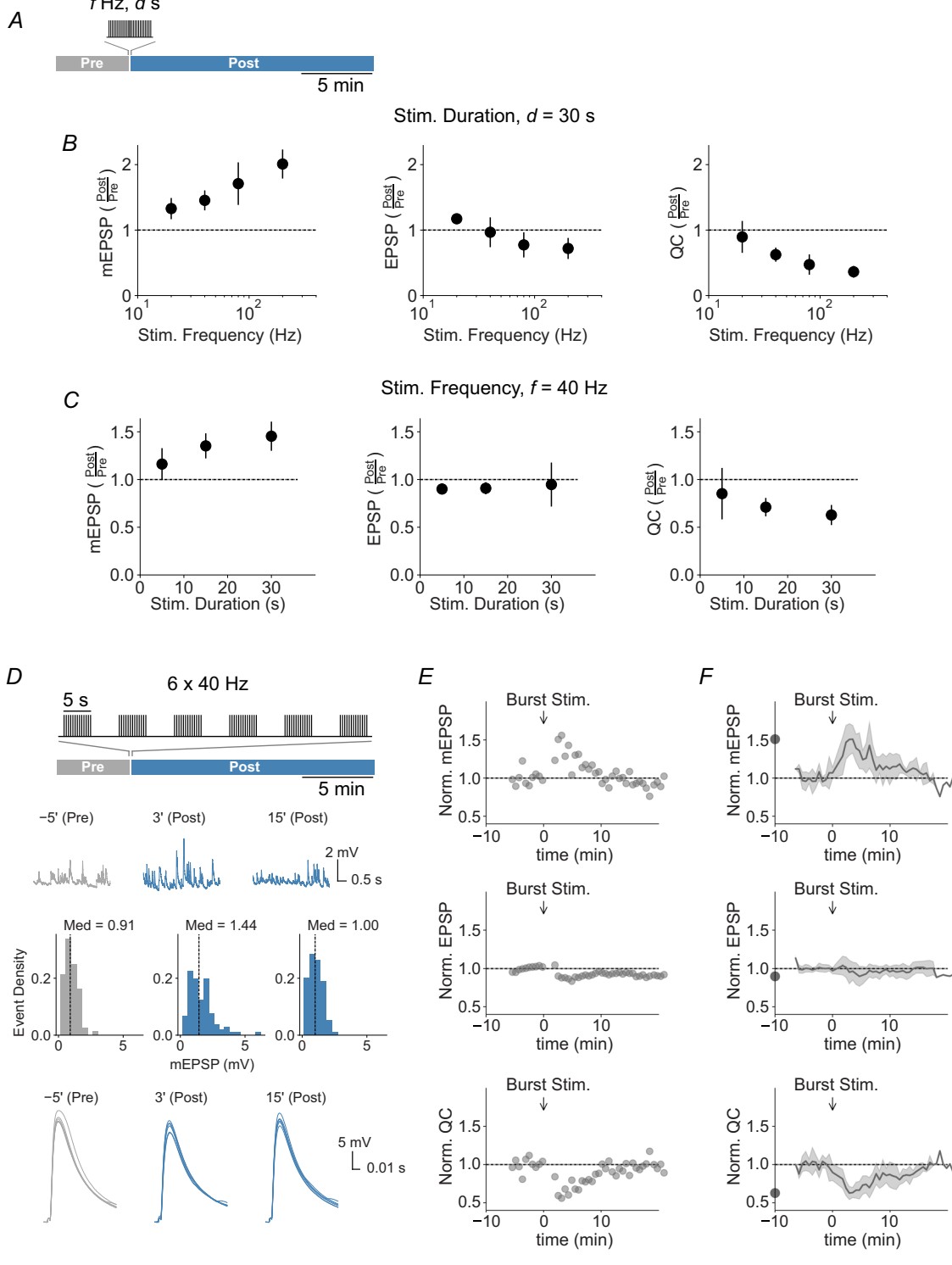

**Figure 2. Post-tetanic quantal size enhancement and quantal content reduction is activity-dependent and occurs upon burst-like stimulation**
*A*, experimental design: mEPSPs and EPSPs (0.2 Hz) were recorded before (Pre) and after (Post) tetanic stimulation of varying frequency (*f*) or duration (*d*). *B*, mean ± SD post-tetanic fractional change (maximum or minimum) of mEPSP amplitude, EPSP amplitude and quantal content (QC) as a function of stimulation frequency, *f*, (train duration: 30 s; *n* = 5, 8, 9 and 8 for stimulation frequency of 20, 40, 80 and 200 Hz, respectively). *C*, mean ± SD post-tetanic fractional change (maximum or minimum) of mEPSP amplitude, EPSP amplitude and quantal content as a function

of stimulation duration (*d*) at a frequency (*f*) of 40 Hz (*n* = 6, 6 and 8 for stimulation durations of 5, 15 and 30 s, respectively). *D*, experimental design (as in *A*) of burst-like stimulation (6 × 40 Hz for 5 s, 5 s interburst interval, top). Representative mEPSPs (second row), mEPSP amplitude densities (along with median value, Med; third row) and EPSP traces (fourth row) 5 min before (−5′), 3 min after (3′) and 15 min after (15′) the tetanic stimulation for a representative cell. *E*, mEPSP amplitudes, EPSP amplitudes and quantal contents (QC = EPSP/mEPSP) from a representative NMJ normalized to pre-stimulation levels (quantification bin width: 35 s) as a function of time relative to the onset of tetanic stimulation. Arrows indicate the timing of burst stimulation. *F*, corresponding mean ± SD values for all recorded NMJs (*n* = 7). The dots on the *y*-axis correspond to the maximum mean post-tetanic change.

## Post-tetanic quantal parameter modulation can be deduced from AP-evoked synaptic transmission

While the prevailing model of synaptic transmission posits that quantal content is defined by dividing EPSP by mEPSP amplitude (del Castillo & Katz, 1954), some evidence suggests a partial separation between AP-evoked and spontaneous miniature synaptic transmission (Fredj & Burrone, 2009; Kavalali, 2015; Melom et al., 2013; Peled et al., 2014). In such a scenario, our data could be interpreted as tetanic stimulation affecting only spontaneous transmission, and not AP-evoked transmission, rendering the observed quantal content changes spurious. To test this possibility, we estimated quantal parameters independent of spontaneous transmission by amplitude fluctuation analysis of AP-evoked EPSCs under conditions of low release probability (extracellular $Ca^{2+}$ concentration of 0.25 mM) using two-electrode voltage clamp (TEVC; see Methods) (McLachlan, 1978). Although tetanic stimulation (80 Hz, 30 s) did not induce significant changes in mean EPSC amplitude ($\mu$) ($P = 0.138$, $n = 15$) (Fig. 3*B*), we observed an increase in EPSC amplitude variance ($\sigma^2$) post-tetanus ($P = 0.0377$) (Fig. 3*A* and *C*). This increase in EPSC amplitude variance without major changes in mean EPSC amplitude results in a reduced squared reciprocal of the coefficient of variance ($1/CV^2 = \mu^2/\sigma^2$; $P = 0.00169$) (Fig. 3*D*), which is an estimate for quantal content at low release probability (McLachlan, 1978). Furthermore, the post-tetanic increase in EPSC amplitude variance with largely unchanged mean EPSC amplitude also indicates an increase in quantal size (quantal size, q $= \sigma^2/\mu$; $P = 0.0067$) (Fig. 3*E*). Quantal content estimated from EPSC amplitude variance correlated well with quantal content calculated by the ratio of EPSC over mEPSC amplitude, and the effect of tetanic stimulation was also correlated between the two estimates (Fig. 3*F*). On the other hand, EPSC mean and variance remained stable in the absence of tetanic stimulation (mean: $P = 0.95$; variance: $P = 0.866$; $1/CV^2$: $P = 0.568$; $\sigma^2/\mu$: $P = 0.742$, $n = 10$) (Fig. 3*G*–*L*). Together, these data provide independent evidence that tetanic stimulation increases quantal size and decreases quantal content. Moreover, this conclusion can be inferred solely from AP-evoked synaptic transmission, suggesting that the increase in quantal size contributes to evoked synaptic transmission after tetanic stimulation.

## Tetanic stimulation decreases readily-releasable pool (RRP) size and synaptic vesicle density

The post-tetanic decrease in quantal content indicates a reduction in neurotransmitter release (Figs 1–3). One potential cause underlying this post-tetanic release depression could be a partial depletion of the pool of release-ready synaptic vesicles. To test this possibility, we estimated RRP size using cumulative EPSC amplitude analysis after short stimulus trains in TEVC (Fig. 4*A* and *B*) (see Methods) (Schneggenburger et al., 1999). Following tetanic stimulation, we observed a decrease in cumulative EPSC amplitude by ∼40% ($P = 0.0115$; $n = 6$) (Fig. 4*C* and *D*), implying a prominent reduction in RRP size. Taking into account the post-tetanic increase in quantal size (Figs 1–3), the observed decrease in cumulative EPSC amplitude is an upper bound for the reduction in RRP size. Normalizing the decrease in cumulative EPSC amplitude to the average post-tetanic increase in mEPSP amplitude (∼75%) indicates that the post-tetanic RRP size could be as low as ∼40% of pre-tetanic levels.

Next, we explored whether the post-tetanic RRP size decrease is accompanied by a decrease in synaptic vesicle density. We addressed this question by visualizing synaptic vesicles and their distribution using transmission electron microscopy (see Methods). Larvae were fixed 3 min after tetanic stimulation for subsequent EM processing, and the density of synaptic vesicles was quantified within a 300 nm radius of the T-bar centre of individual active zones (see Methods) (Fig. 4*E*). Our analysis revealed a post-tetanic decrease in synaptic vesicle density by ∼21% compared to control boutons ($P = 0.0052$; $n = 21$ and $19$) (Fig. 4*G*). Together, these data show that the post-tetanic decrease in quantal content is accompanied by a decrease in RRP size and synaptic vesicle density. Moreover, these observations suggest that the resupply of synaptic vesicles is probably saturated under our experimental conditions.

## No evidence for a role of postsynaptic GluR modulation and $Ca^{2+}$ signalling in post-tetanic quantal size enhancement

High-frequency stimulation not only reduces quantal content and RRP size, but also increases quantal size (Figs 1–3). We next investigated the observed quantal size

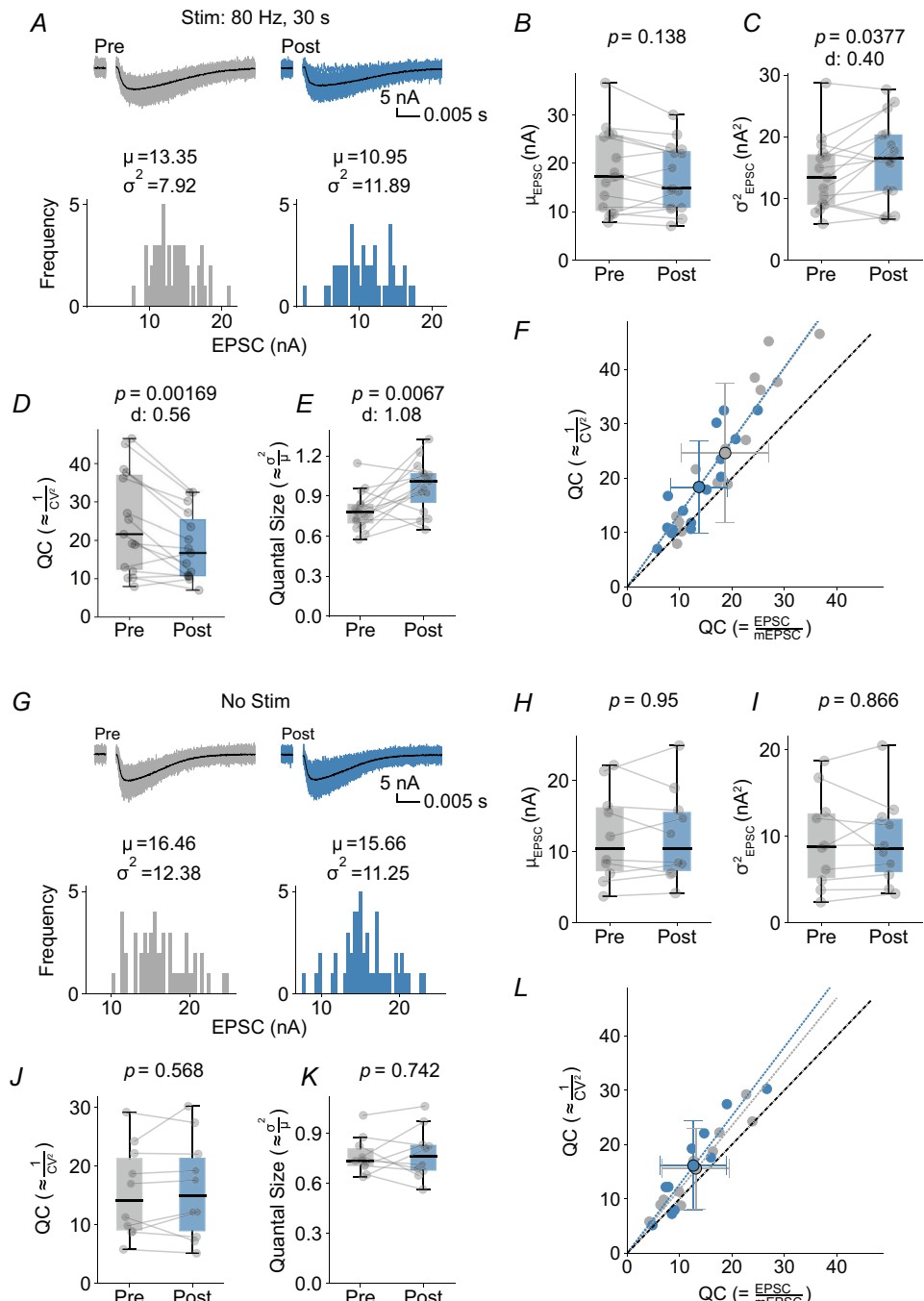

**Figure 3. EPSC fluctuation analysis supports post-tetanic increase in quantal size and decrease in quantal content**

*A*, representative EPSCs (50 traces per group, averages are shown in black) traces and amplitude distributions before (Pre, grey) and 3 min after (Post, blue) tetanic stimulation (80 Hz for 30 s). *B*, mean EPSC amplitude ($\mu$; $P = 0.138$) and (*C*) EPSC amplitude variance ($\sigma^2$; $P = 0.0377$) before and after tetanic stimulation ($n = 15$). *D*, the inverse of the squared coefficient of variation ($1/CV^2$, $P = 0.00169$) of EPSC amplitudes as an estimate for quantal content (QC). (*E*) $\sigma^2/\mu$ as an estimate for quantal size before and after tetanic stimulation ($P = 0.0067$). *F*, quantal content estimated from EPSC fluctuation analysis ($1/CV^2$) *vs.* quantal content estimated as the ratio of EPSC and mEPSC before (grey points) and after (blue points) tetanic stimulation. Data points with error bars represent the mean ± SD of the respective group. The dashed colored and black lines represent line fits to the respective groups and a unity line, respectively. Quantal content estimates from both methods are well-correlated. *G*, representative EPSC traces (50 traces per group, averages are shown in black) and amplitude distributions of an

unstimulated control NMJ (No Stim). Analysis time windows were matched with the stimulated group. *H*, mean EPSC amplitude ($\mu$; *P* = 0.95) and (*I*) EPSC amplitude variance ($\sigma^2$; *P* = 0.866) (*n* = 10). *J*, inverse squared coefficient of variation (1/CV$^2$; *P* = 0.568) and (*K*) $\sigma^2/\mu$ (*P* = 0.742) of EPSC amplitudes. *L*, quantal content estimated from EPSC fluctuation analysis (1/CV$^2$) *vs.* quantal content estimated from EPSP/mEPSPs for unstimulated controls (as in *F*).

increase after high-frequency stimulation because it may confer robustness to AP-evoked synaptic transmission after repetitive stimulation. An increase in quantal size is mostly associated with postsynaptic mechanisms affecting neurotransmitter receptors (DiAntonio et al., 1999; Nusser et al., 1997; O'Brien et al., 1998; Turrigiano et al., 1998). We therefore next probed postsynaptic ionotropic glutamate receptors (iGluRs) after tetanic stimulation. iGluRs at the *Drosophila* NMJ assemble as tetramers comprising the essential subunits GluRIIC, D and E (Featherstone et al., 2005; Marrus & DiAntonio, 2004; Petersen et al., 1997; Qin et al., 2005; Schuster et al., 1991), along with either a GluRIIA or a GluRIIB subunit (Petersen et al., 1997; Schuster et al., 1991). We first compared the fluorescence intensity of antibodies targeting GluRIIA and GluRIIC between unstimulated and tetanically stimulated NMJs. We did not observe apparent differences in GluRIIA (*P* = 0.965) or GluRIIC (*P* = 0.765) fluorescence intensity between unstimulated (Ctrl: *n* = 6) and stimulated (Stim: *n* = 6) NMJs (Fig. 5*A*), suggesting no major changes in iGluR abundance or subunit composition. We next investigated whether the post-tetanic quantal size increase depends on GluR sub-unit composition by monitoring mEPSPs in null mutants for either *GluRIIA* (*GluRIIA^SP16^*) (Petersen et al., 1997) or *GluRIIB* (*GluRIIB^SP5^*) (Muttathukunnel et al., 2022). Both *GluRIIA^SP16^* and *GluRIIB^SP5^* mutant NMJs exhibited a prominent post-tetanic increase in mEPSP amplitude (*GluRIIA*: *P* < 0.001; *n* = 5; *GluRIIB*: *P* < 0.001; *n* = 8) (Fig. 5*B* and *C*), again implying no major involvement for iGluR subunit-specific modulation in post-tetanic quantal size modulation.

Another possible mechanism that may underlie the post-tetanic quantal size increase is the accumulation of postsynaptic intracellular Ca$^{2+}$. This accumulation may trigger Ca$^{2+}$-dependent signalling, including post-translational iGluR modification, as observed for mammalian receptors (Mao et al., 2011). The major sources of postsynaptic Ca$^{2+}$ at the *Drosophila* NMJ are iGluRs, L-type Ca$^{2+}$ channels and intracellular stores (Chang et al., 1994; Gielow et al., 1995; Hasan & Venkiteswaran, 2010). Ca$^{2+}$-impermeable iGluR mutant NMJs (*GluRIIA^Q615R^*, *GluRIIB^SP6^*) (Han et al., 2023) incubated with the L-type Ca$^{2+}$ channel blocker nifedipine and the ryanodine receptor blocker ryanodine displayed a prominent increase in mEPSP amplitude after tetanic stimulation (*P* < 0.001; *n* = 6) (Fig. 5*D*), suggesting that postsynaptic Ca$^{2+}$ does not significantly contribute to the post-tetanic quantal size increase.

It is conceivable that high-frequency activation of iGluRs increases quantal size via other receptor-mediated signalling mechanisms. We tested this hypothesis by directly stimulating postsynaptic receptors using a high-frequency train of iontophoretic glutamate puffs (see Methods), thereby bypassing release from the pre-synaptic motoneuron. Tetanic stimulation in the form of glutamate puffs produced no increase in mEPSP amplitudes (*P* = 0.920; *n* = 8) (Fig. 5*E*), indicating that the quantal size increase observed after high-frequency stimulation is probably not triggered by postsynaptic GluR-mediated signalling. Additionally, it highlights the possibility that the post-tetanic increase in quantal size may have a presynaptic origin.

## Post-tetanic quantal size enhancement and EPSC maintenance require vesicle recycling

We next probed the role of presynaptic processes in the post-tetanic quantal size increase. Given that postsynaptic receptors are not saturated by the release from individual synaptic vesicles at the *Drosophila* NMJ (Daniels et al., 2004), an increase in vesicular neurotransmitter content could contribute to the post-tetanic quantal size increase. Synaptic vesicle recycling is critical to sustain high-frequency synaptic activity (Stevens & Wesseling, 1998; Wu et al., 2005). A possible hypothesis is thus that tetanic stimulation results in changes in the vesicle recycling process that leads to larger quantal size. To test this possibility, we perturbed the synaptic vesicle cycle by inhibiting dynamin, a small GTPase essential for synaptic vesicle endocytosis (Takei et al., 1995). Application of the dynamin inhibitor dynasore (Macia et al., 2006) (see Methods) attenuated the post-tetanic mEPSP increase to ~25% compared to ~80% in DMSO-treated controls (*P* < 0.001; *n* = 9 and 7) (Fig. 6*A* and *B*), suggesting an involvement of dynamin-mediated endocytosis. By contrast, the post-tetanic increase in mEPSP frequency was not affected by dynasore treatment (*P* = 0.992; *n* = 9 and 7) (Fig. 6*B*), implying that separable mechanisms promote the post-tetanic increase in miniature amplitude and frequency. We then hypothesized that the post-tetanic quantal size increase may compensate for the quantal content reduction, thereby stabilizing AP-evoked synaptic transmission (Figs 1–3). In line with this hypothesis, we observed a stronger post-tetanic decrease in EPSP amplitude in the presence of dynasore (*P* = 0.0484) (Fig. 6*A* and *B*). The post-tetanic quantal content reduction was similar between dynasore-treated and

control NMJs ($P = 0.168$) (Fig. 6*A* and *B*), suggesting that the post-tetanic EPSP amplitude reduction was not the result of a direct effect of dynasore on quantal content reduction. These data imply that the post-tetanic reduction in EPSP amplitude in the presence of dynasore is the result of a reduced post-tetanic quantal size increase.

We confirmed the presynaptic role of dynamin by presynaptic overexpression of the temperature-sensitive dominant negative dynamin transgene *shibire^ts* (*OK371-Gal4>UAS-shibire^ts*) (Kitamoto, 2001) (see Methods). At the restrictive temperature (30°C), NMJs expressing presynaptic *shibire^ts* exhibited a lower post-tetanic quantal size increase of ~25% compared to ~70% in the temperature-matched control (*UAS-shibire^ts/+*) ($P = 0.003$; $n = 8$ and 7) (Fig. 6*C* and *D*). Similar to dynasore treatment, the

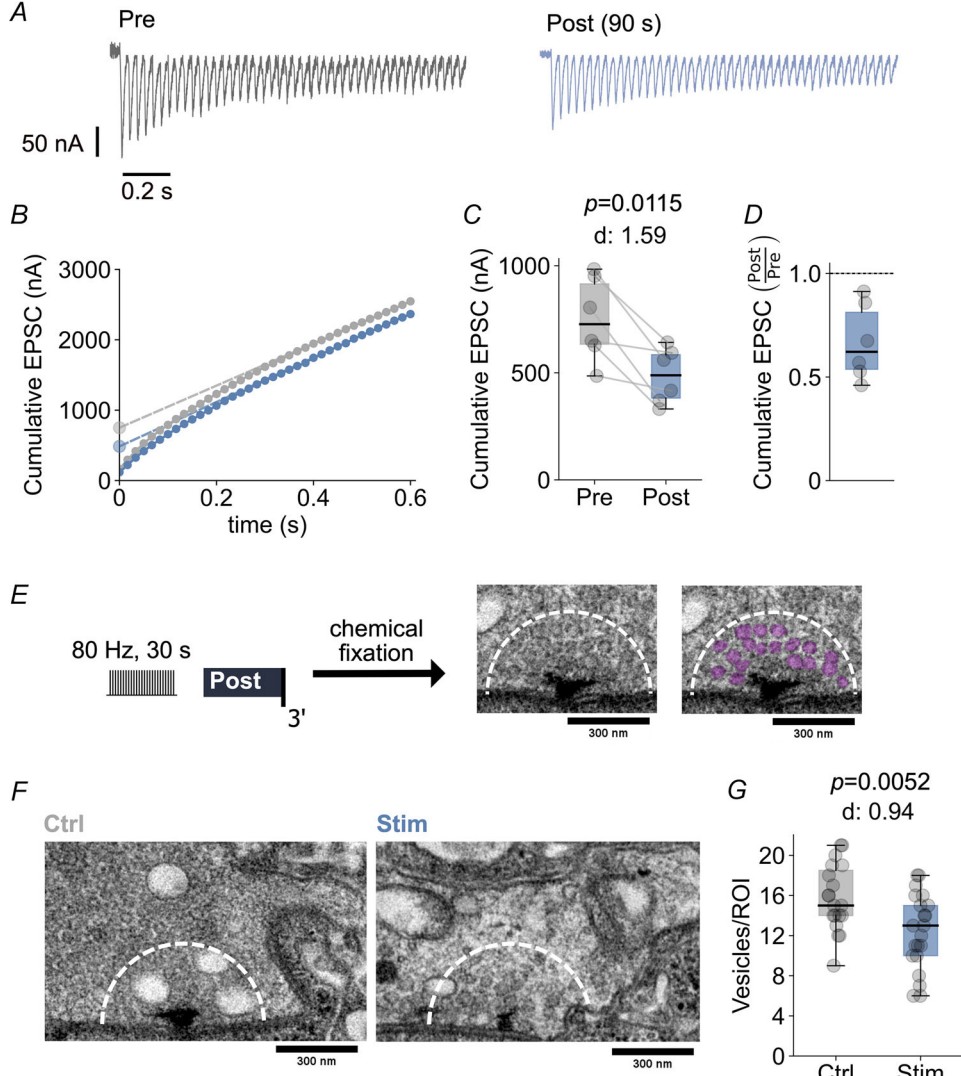

**Figure 4. Reduced RRP size and synaptic vesicle density after tetanic stimulation**
*A*, representative EPSC traces and (*B*) cumulative EPSC amplitudes for a 60-Hz train before (Pre, grey) and 90 s after (Post, blue) tetanic stimulation (80 Hz, 30 s). Line fits to the last 10 data points (coloured dashed lines) were back-extrapolated to the first EPSC (dot at *y*-intercept) for cumulative EPSC amplitude analysis. *C*, cumulative EPSC amplitudes before and after the tetanus (80 Hz, 30 s) for $n = 6$ NMJs. Cumulative EPSC decreased after tetanus stimulation ($P = 0.0115$). *D*, fractional reduction in cumulative EPSC amplitude after tetanic stimulation, presented as the ratio of Post to Pre estimates from (*C*). *E*, experimental design: Larvae were fixed 3 min after tetanic stimulation (80 Hz, 30 s) for transmission electron microscopy. Vesicles were segmented manually (magenta) within a 300 nm radius around the T-bar centre. *F*, representative electron micrographs of an unstimulated (Ctrl) and a stimulated (Stim) NMJ. For visualization purposes, the contrast of the raw images was enhanced using Fiji. *G*, quantification of vesicle density for unstimulated and stimulated NMJs. Tetanic stimulation reduced synaptic vesicle density ($P = 0.0052$; $n = 19$ and 21). Note that synaptic vesicle densities are probably underestimated because segmentation was confined to clearly discernable vesicles to reduce false positive segmentation errors.

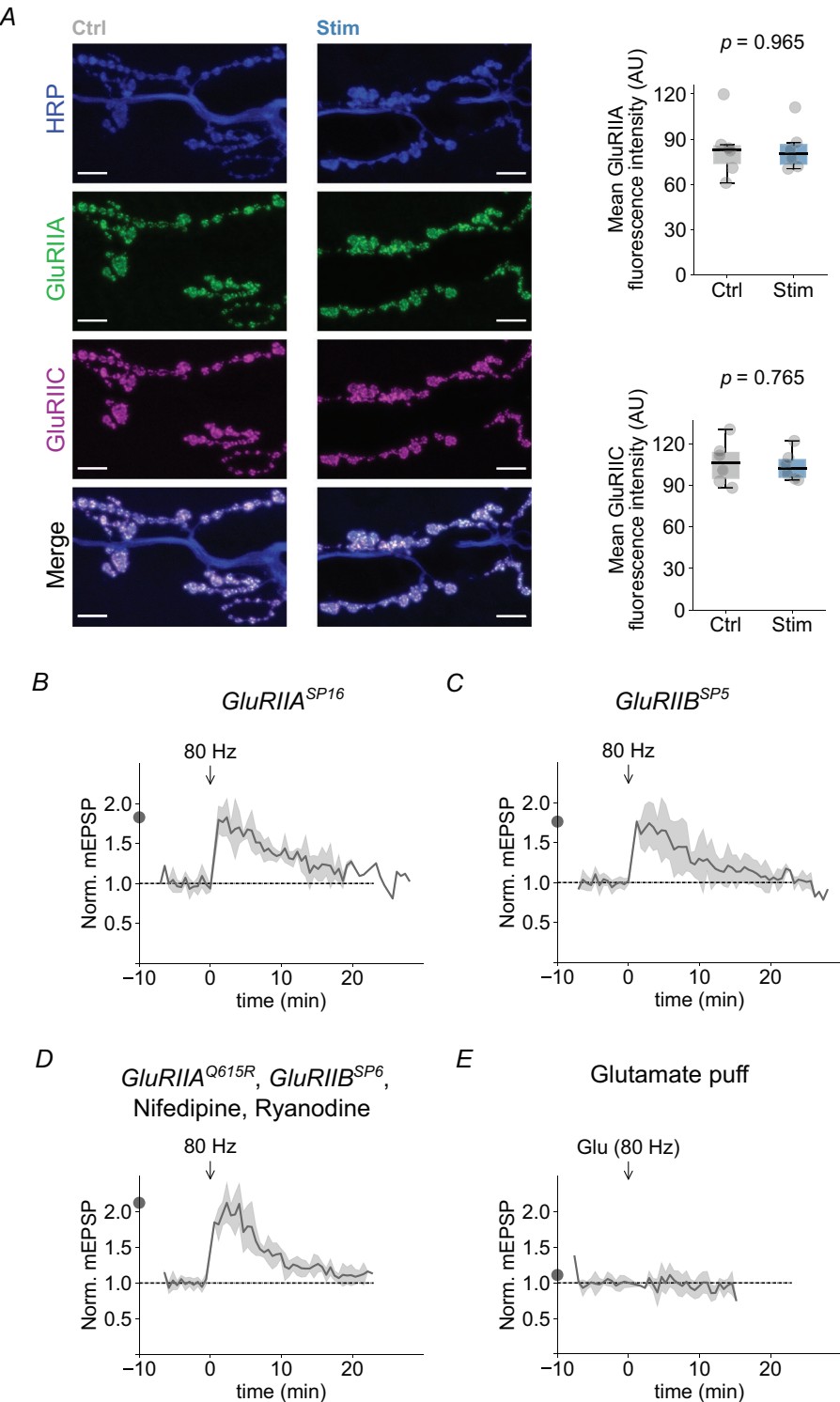

**Figure 5. No apparent role for postsynaptic GluR modulation or Ca²⁺ signalling in post-tetanic quantal size enhancement**

*A*, left: representative maximum intensity *z*-projection confocal images of the neuronal membrane marker HRP (top), GluRIIA (second row) and GluRIIC (third row) of an NMJ without (Ctrl) and another NMJ with (Stim) tetanic stimulation (80 Hz, 30 s). Scale-bar, 10 μm. Right: mean GluRIIA (*P* = 0.965) and GluRIIC (*P* = 0.765) fluorescence intensities without or with tetanic presynaptic stimulation (80 Hz, 30 s; *n* = 6 per group). mEPSP

amplitude (mean ± SD) normalized to the respective pre-tetanic mean values of (B) GluRIIA$^{SP16}$, (C) GluRIIB$^{SP5}$ and (D) GluRIIA$^{Q615R}$, GluRIIB$^{SP6}$ mutant NMJs in the presence of nifedipine and ryanodine ($n$ = 5, 8 and 6 for B–D, respectively). E, mEPSP amplitudes (mean ± SD in response to high-frequency iontophoretic glutamate puffs (80 Hz, 30 s) normalized to the respective pre-tetanic mean values from a wild-type NMJ ($n$ = 8). Arrows indicate the timing of stimulation, and data are plotted as a function of time relative to the onset of tetanic stimulation. The dots on the $y$-axis correspond to the maximum mean post-tetanic change.

post-tetanic increase in mEPSP frequency persisted in *shibire$^{ts}$* mutants ($P$ = 0.646; $n$ = 8 and 7) (Fig. 6D). We also observed a stronger post-tetanic reduction in EPSP amplitudes at *shibire$^{ts}$* expressing NMJs, compared to controls ($P$ < 0.001) (Fig. 6C and D). However, the latter observation may be confounded by a stronger post-tetanic quantal content reduction at *shibire$^{ts}$*-expressing NMJs ($P$ = 0.002) (Fig. 6D). Together, these findings provide evidence for a critical involvement of presynaptic dynamin and vesicle recycling in post-tetanic quantal size enhancement and the maintenance of AP-evoked EPSP amplitude. By extension, our data imply that separable mechanisms mediate the post-tetanic increase in quantal amplitude and frequency.

### Post-tetanic quantal size enhancement and EPSC maintenance require vesicle refilling

Following endocytosis, H$^+$-ATPase-dependent acidification of synaptic vesicles is required for neurotransmitter filling (Edwards, 2007). Similar to the effects of interfering with dynamin, blocking vesicular H$^+$-ATPase activity with bafilomycin also diminished the post-tetanic mEPSP amplitude increase compared to DMSO-treated controls ($P$ < 0.001; $n$ = 6 and 9), with minor effects on the post-tetanic decrease in quantal content ($P$ = 0.0526), thereby preventing EPSP amplitude maintenance ($P$ = 0.0181) after tetanic stimulation (Fig. 7). Of note, we cannot rule out that the decrease in baseline EPSP amplitude prior to tetanic stimulation at bafilomycin-treated NMJs (Fig. 7B) contributes to the post-tetanic decrease in EPSP amplitude and quantal content (Fig. 7B). Again, bafilomycin treatment did not affect the post-tetanic increase in mEPSP frequency ($P$ = 0.883) (Fig. 7B). These data support the idea that synaptic vesicle recycling is required for post-tetanic quantal size enhancement and the maintenance of AP-evoked EPSP amplitude.

### Tetanic stimulation increases synaptic vesicle diameter

Larger synaptic vesicles could result in an increased quantal size (Daniels et al., 2004; Dickman et al., 2005; Karunanithi et al., 2002; Steinert et al., 2006; Zhang et al., 1998). A post-tetanic increase in synaptic vesicle size with higher neurotransmitter content could thus contribute to the quantal size increase observed here. To test whether tetanic stimulation increases synaptic vesicle size, we employed transmission electron microscopy (see Methods) and quantified synaptic vesicle diameters within a 300 nm radius of the T-bar centre (Fig. 8A). This analysis revealed a right shift in the distribution of synaptic vesicle outer diameter after tetanic stimulation ($P$ < 0.001; $n$ = 19 and 21) (Fig. 8B and C). We detected a similar right shift in the synaptic vesicle outer diameter distribution after tetanic stimulation in a region of interest outside of active zones ($P$ < 0.001; $n$ = 19 and 21) (see Methods) (Fig. 8D–F), suggesting bouton-wide changes in synaptic vesicle diameter. Consistent with previous observations (Akbergenova & Bykhovskaia, 2009), we also noted larger vesicular structures resembling endocytic structures at stimulated NMJs (Fig. 6A, right), which were not considered in our analysis (see Methods). Considering the thickness of the synaptic vesicle membrane at the *Drosophila* NMJ (Daniels et al., 2004), the observed increase in mean synaptic vesicle outer diameter (∼10%) translates into a prominent increase in synaptic vesicle volume (∼70%; see Discussion). Thus, an enlargement of synaptic vesicles in response to tetanic stimulation could be a potential mechanism for the post-tetanic quantal size increase observed in the present study.

### Discussion

In the present study, we have uncovered a post-tetanic increase in quantal size that counteracts post-tetanic release depression within minutes after tetanic stimulation at the *Drosophila* NMJ. Tetanic stimulation decreased presynaptic release and synaptic vesicle density, whereas evoked amplitudes were largely unchanged. At the same time, quantal size and synaptic vesicle diameter were increased after the tetanus. Our data suggest that this increase in quantal size is presynaptically driven, as evidenced by its persistence despite genetic or pharmacological manipulations targeting postsynaptic glutamate receptors or calcium channels. Instead, perturbations of synaptic vesicle recycling attenuated both, the post-tetanic quantal size increase and the maintenance of evoked responses. Together, our data are consistent with the idea that high-frequency synaptic activity results in a presynaptically-driven increase in quantal size that counteracts release depression, thereby stabilizing synaptic efficacy after tetanic stimulation.

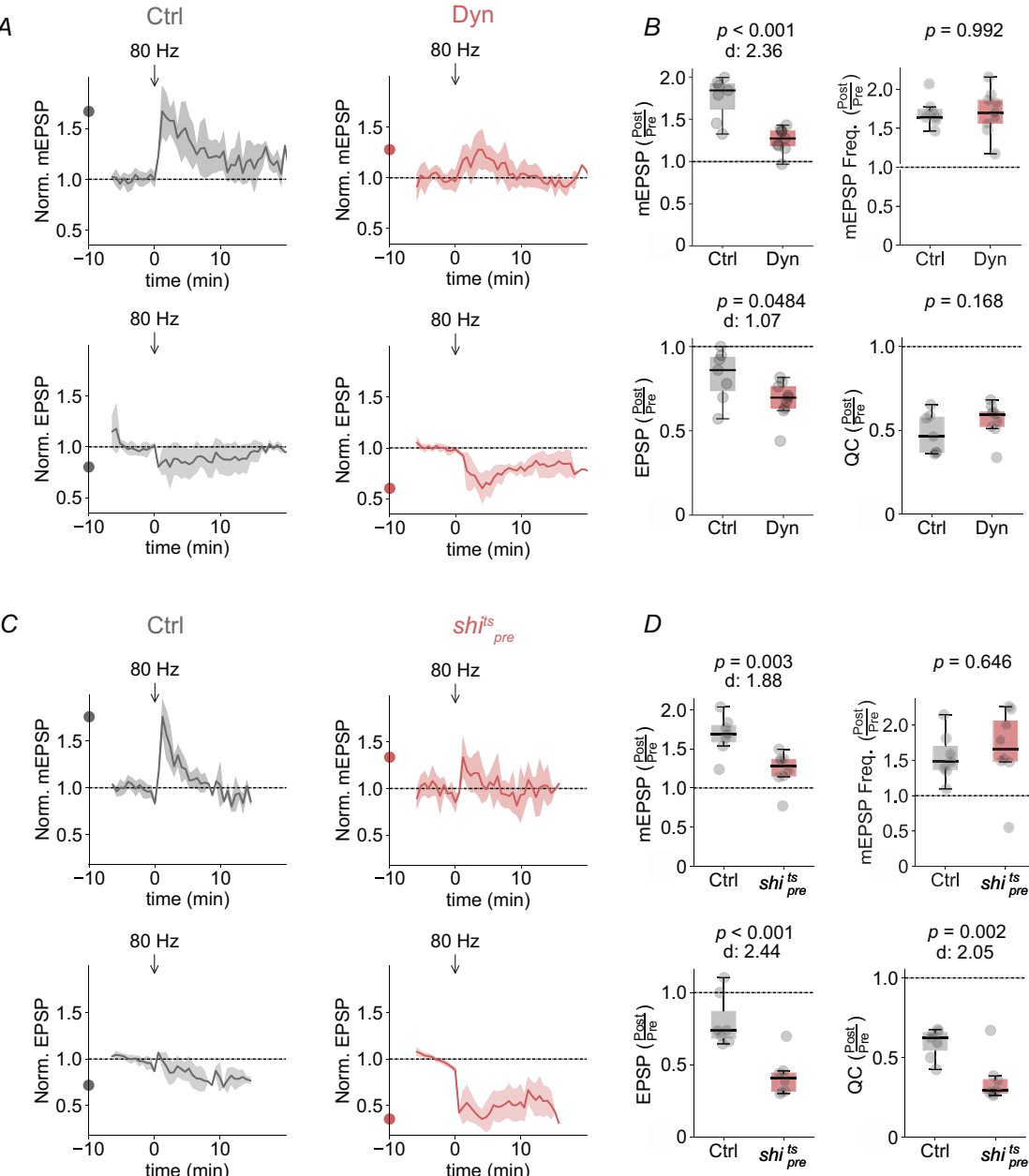

**Figure 6. Impeding dynamin-dependent vesicle recycling attenuates post-tetanic quantal size enhancement and EPSC maintenance**

*A*, mean mEPSP amplitude and EPSP amplitude values averaged for 35 s bins and normalized to the mean pre-tetanic levels in the presence of DMSO (Ctrl) or dynasore (Dyn) (*n* = 7 and 9, respectively). Arrows indicate the timing of tetanic stimulation (80 Hz, 30 s) and data are plotted as a function of time relative to the onset of tetanic stimulation. *B*, quantification of maximal fractional changes in mEPSP amplitude (*P* < 0.001), mEPSP frequency (*P* = 0.992), EPSP amplitude (*P* = 0.0484) and quantal content (*P* = 0.168) normalized to the respective mean pre-tetanic values (Post/Pre). *C*, mean mEPSP amplitude and EPSP amplitude values averaged for 35 s bins and normalized to the mean pre-tetanic levels in control *UAS-shi*ts/+, (Ctrl; 30°C) and NMJs expressing presynaptic *shibire*ts, *OK371-Gal4/UAS-shi*ts (*shi*ts_pre; 30°C) (*n* = 7 and 8, respectively). *D*, quantification of mEPSP amplitude (*P* = 0.003), mEPSP frequency (*P* = 0.646), EPSP amplitude (*P* < 0.001) and quantal content (*P* = 0.002) fractional changes in control *vs. shi*ts_pre. The dots on the *y*-axis of time course traces correspond to the maximum mean post-tetanic change.

Several observations indicate release depression after tetanic stimulation in this study. First, the increase in mEPSP amplitude in combination with the maintenance of evoked EPSP amplitudes translate into a decrease in quantal content (Figs 1 and 2). Second, EPSC amplitude fluctuation analysis independently confirmed the decrease in quantal content (Fig. 3). Third, tetanic stimulation decreased cumulative EPSC amplitude, consistent with a decrease in RRP size and release (Fig. 4). Fourth, dynamin and $H^+$-ATPase perturbation revealed a post-tetanic decrease in EPSP amplitudes (Figs 6 and 7). Furthermore, synaptic vesicle density was decreased following tetanic stimulation (Fig. 4), suggesting a reduction in synaptic vesicle number, which probably contributes to release depression. The post-tetanic decrease in RRP size was more pronounced than the decrease in synaptic vesicle density (Fig. 4), implying that saturation of synaptic vesicle resupply is probably not the only factor under-ling post-tetanic release depression. Based on previous work (Neher, 2010), the limited availability of functional release sites during sustained activity may contribute to the post-tetanic decrease in quantal content observed here.

At the *Drosophila* NMJ, a significant fraction (∼30% to ∼85%) of all synaptic vesicles are thought to reside in a reserve pool, which does not undergo synaptic vesicle recycling upon low-frequency stimulation (Delgado et al., 2000; Poskanzer & Davis, 2004). Our analysis cannot differentiate whether the vesicles released after tetanic stimulation are drawn from a recycling or a reserve pool. However, tetanic stimulation has been shown to recruit synaptic vesicles from the reserve pool, and newly endocytosed vesicles are recycled into both pools upon tetanic stimulation at the *Drosophila* NMJ (Kuromi & Kidokoro, 1998; Kuromi & Kidokoro, 2000). Intriguingly, the recruitment of reserve pool vesicles continues after tetanic stimulation (Kuromi & Kidokoro, 2000). Hence, reserve pool vesicles probably contribute to release after the tetanic stimuation protocol used in the present study. There is evidence that reserve pool vesicles predominately locate towards the bouton centre at the *Drosophila* NMJ (Kuromi & Kidokoro, 2000). Assuming that the larger synaptic vesicles seen after tetanic stimulation are newly endocytosed, the bouton-wide changes in synaptic vesicle diameter (Fig. 8) indicate that a significant fraction of vesicles, including reserve pool vesicles, were recycled after the tetanus. Furthermore, considering the total number of synaptic vesicles per NMJ (∼84,000) (Atwood et al., 1993, Kuromi & Kidokoro, 2003) and release-ready vesicle resupply rates during sustained activity per NMJ (∼1000–7500 vesicles $s^{-1}$) (Hallermann et al., 2010; Delgado et al., 2000), our tetanic stimulation protocol is

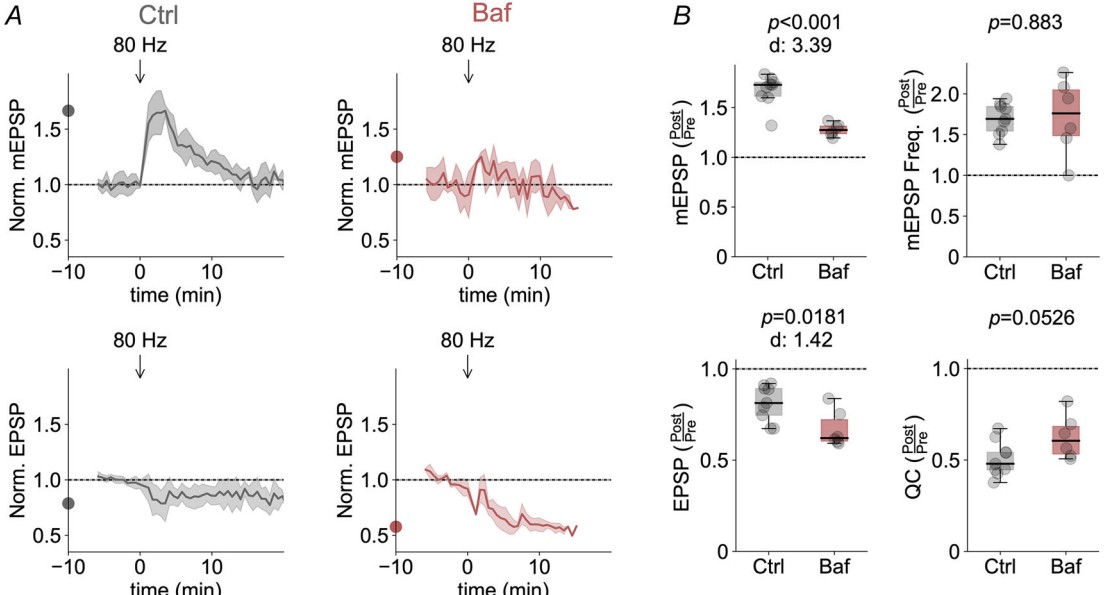

**Figure 7. Impeding vesicular $H^+$-ATPase activity attenuates post-tetanic quantal size enhancement and EPSC maintenance**
*A*, mean mEPSP amplitude and EPSP amplitude values averaged for 35 s bins and normalized to the mean pre-tetanic levels in the presence of DMSO (Ctrl) or bafilomycin (Baf) (*n* = 7 and 9, respectively) as a function of time relative to the onset of tetanic stimulation. Arrows indicate the timing of tetanic stimulation (80 Hz, 30 s). *B*, quantification of maximal fractional changes in mEPSP amplitude (*P* < 0.001), mEPSP frequency (*P* = 0.883), EPSP amplitude (*P* = 0.0181) and quantal content (*P* = 0.0526) normalized to the respective mean pre-tetanic values (Post/Pre). The dots on the *y*-axis of the time course traces correspond to the maximum mean post-tetanic change.

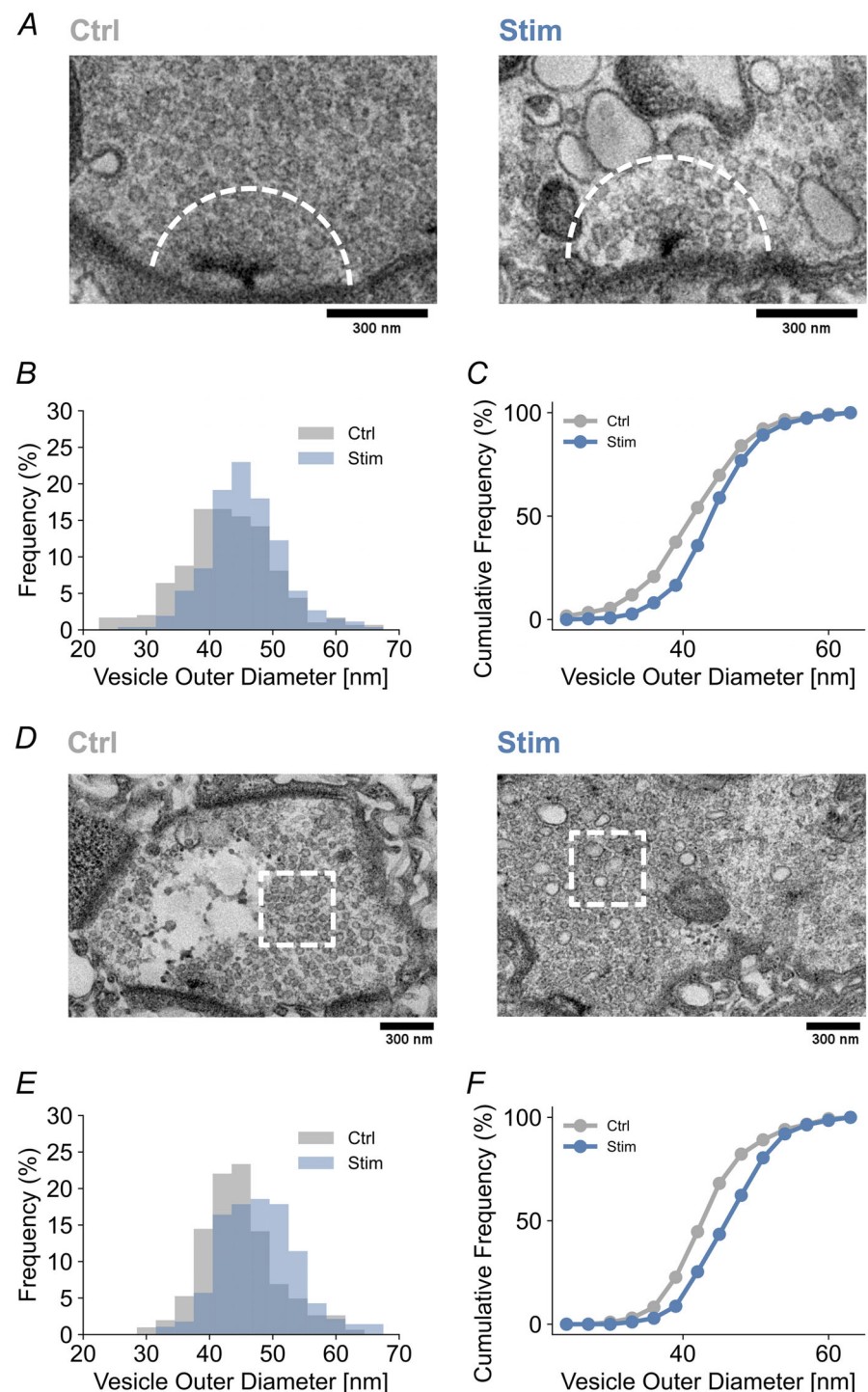

**Figure 8. Increased synaptic vesicle diameter upon tetanic stimulation**
*A*, representative electron micrographs of a control (Ctrl) and a stimulated (Stim; 80 Hz, 30 s) NMJ (Experimental procedure as in Fig. 4*E*). Vesicles were analyzed within a 300 nm radius around the T-bar centre (white dashed line indicates radius). *B*, relative frequency distribution histogram of vesicle outer diameter at control and stimulated active zones. *C*, cumulative frequency distribution of vesicle outer diameter of control and stimulated active zones (bin width: 3 nm; range 24–67 nm; *P* < 0.001; Kolmogorov–Smirnov test). *D*, representative electron micrographs of a control and a stimulated synaptic bouton. Vesicles were analyzed within 400 nm × 400 nm regions placed outside of active zones, towards bouton centres (white dashed squares). *E*, relative frequency distribution histogram of vesicle outer diameter at control and stimulated boutons. *F*, cumulative frequency distribution of vesicle outer diameter of control and stimulated boutons (bin width: 3 nm; range 24–67 nm; *P* < 0.001; Kolmogorov–Smirnov test). The contrast of the representative electron micrographs was enhanced using Fiji.

expected to trigger the fusion of a large fraction of the total synaptic vesicle pool. Together, although we cannot differentiate between the post-tetanic release of recycling and reserve pool vesicles, our protocol probably recruited recycled vesicles from both pools.

At many synapses, tetanic stimulation potentiates evoked responses as a result of a net increase in release probability and RRP size defined as PTP (Delaney et al., 1989; Korogod et al., 2005; Zucker & Regehr, 2002). There is also some evidence for post-tetanic release depression (Betz, 1970; Guo & Zhong, 2006; Neale et al., 2001; Storozhuk et al., 2002; von Gersdorff et al., 1997). In contrast to these previous reports, in which tetanic stimulation induced a net change in evoked amplitudes, EPSCs remained rather stable within a broad activity range in the present study (Fig. 2). The concomitant attenuation of post-tetanic quantal size potentiation, and the increased depression of evoked responses seen after interfering with synaptic vesicle recycling suggest that post-tetanic release depression is compensated by the post-tetanic quantal size increase. Thus, under conditions of limited release-ready vesicles resupply, an increase in quantal size stabilizes synaptic transmission, thereby masking post-tetanic release depression, and extending the NMJ's dynamic range.

Post-tetanic increases in mEPSP frequency or amplitude have been observed at various synapses (He et al., 2009; Kombian et al., 2000; Quinlan & Hirasawa, 2013; Rudolph et al., 2015). At the *Drosophila* NMJ, tetanic stimulation was shown to increase miniature frequency (Jorquera et al., 2012). By contrast, a post-tetanic increase in miniature amplitude had not been described at the *Drosophila* NMJ. Interestingly, $K^+$-induced depolarization for 5 min of the *Drosophila* NMJ induces a concomitant increase in quantal size and synaptic vesicle size, as well as large endocytic structures (Akbergenova & Bykhovskaia, 2009). Furthermore, an increase in mEPSP amplitude has been reported after high larval crawling activity (Steinert et al., 2006). Considering the high-frequency activity needed for excitation-contraction coupling at the *Drosophila* NMJ (Ormerod et al., 2022), sustained larval locomotor activity could induce similar quantal size increases as the ones described in the present study. What could underlie the post-tetanic increase in quantal size? The modulation of quantal size is generally associated with changes in postsynaptic neurotransmitter receptors (Turrigiano et al., 1998). However, our data support the conclusion that the post-tetanic quantal size increase probably does not involve postsynaptic changes, consistent with studies describing a post-tetanic increase in quantal size attributed to multivesicular release or compound fusion (He et al., 2009; Kombian et al., 2000; Quinlan & Hirasawa, 2013; Rudolph et al., 2015). Multivesicular release and compound fusion typically correlate with miniature frequency. The post-tetanic potentiation of miniature amplitude described here is probably not a result of multivesicular release or compound fusion because dynamin and $H^+$-ATPase perturbation attenuated the post-tetanic increase in miniature amplitude, whereas the increase in miniature frequency persisted. Miniature frequency and release probability are enhanced during PTP (Korogod et al., 2005). The observation of reduced post-tetanic evoked amplitudes, but increased miniature frequency after impeding synaptic vesicle recycling thus suggests that the post-tetanic maintenance of evoked amplitudes is probably not driven by a PTP-like increase in release probability.

The post-tetanic increase in quantal size was paralleled by an increase in synaptic vesicle diameter near active zones, as well as outside of active zones (Fig. 8). Taking into account the thickness of the synaptic vesicle membrane at the *Drosophila* NMJ (∼9 nm) (Daniels et al., 2004), the observed increase in synaptic vesicle outer diameter (∼10%) translates into a ∼70% increase in synaptic vesicle volume. Such a change in vesicle size could be sufficient to produce the observed ∼60–80% increase in mEPSP amplitude, in line with earlier work describing similar changes in synaptic vesicle volume and quantal size upon overexpression of the *Drosophila* vesicular glutamate transporter at *Drosophila* NMJ (Daniels et al., 2004). Hence, the increase in synaptic vesicle size is probably a major cause underlying the post-tetanic increase in quantal size.

What could underlie the post-tetanic increase in synaptic vesicle size? Dynamin perturbation attenuated the post-tetanic increase in vesicle size (Fig. 6). Intriguingly, at the *Drosophila* NMJ, several endocytosis mutants display an increase in synaptic vesicle size and quantal size resembling the post-tetanic changes observed here (Dickman et al., 2005; Koh et al., 2004; Zhang et al., 1998; Zhao et al., 2013). In particular, for mutations in dynamin-associated protein 160 kDa (dap160), these changes are also associated with a prominent reduction in the levels of dynamin, synaptojanin and endophilin (Koh et al., 2004). At the calyx of Held, tetanic stimulation leads to a quantal size increase only in endocytosis mutants (dynamin-1 KO) (Mahapatra & Lou, 2017). These observations highlight the intriguing possibility that synapses enter a potentially maladaptive vesicle recycling regime during high-frequency activity, perhaps as a result of saturation or physical limits of the endocytosis machinery, resulting in larger synaptic vesicles. The resulting increase in quantal size may contribute to post-tetanic potentiation, as shown at the calyx of Held (Mahapatra & Lou, 2017), or in this case, the stabilization of synaptic transmission. Hence, a limitation in synaptic vesicle recycling may be adaptive by stabilizing synaptic efficacy in response to repetitive activity.

The transient increase in quantal size observed in response to tetanic stimulation at the *Drosophila* NMJ presents a previously unappreciated adaptive mechanism through which synapses sustain robust transmission after prolonged repetitive activity. Furthermore, the magnitude of this effect scales with the frequency and duration of synaptic activity, thus enabling synapses to maintain stable synaptic transmission levels over a broad activity range. Importantly, this phenomenon is not limited to tetanic stimulation, but also manifests in response to a burst-like activity pattern (Fig. 2). Similar increases in synaptic vesicle size and quantal size were detected after sustained larval locomotor activity (Steinert et al., 2006), implying potential implications of our findings for animal behaviour. Future work will test whether the form of adaptive synaptic regulation discovered here also confers robustness to other synapses.

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

## Additional information

### Data availability statement

All data supporting the results in the manuscript are included within the figures and the text. Raw data will be freely available from the corresponding author upon reasonable request.

### Competing interests

The authors declare that they have no competing interests.

### Author contributions

A.G.N. and M.M. were responsible for the conception and design of the work. A.G.N., N.B., L.S., J.K. and M.M. were responsible for acquisition, analysis, or interpretation of data for the work. A.G.N., N.B., L.S., J.K., J.M.M.M. and M.M. were responsible for drafting the work or revising it critically for important intellectual content. All authors have approved the final version of the manuscript submitted for publication and agree to be accountable for all aspects of the work in ensuring that questions related to the accuracy or integrity of any part of the work are appropriately investigated and resolved. All persons designated as authors qualify for authorship, and all those who qualify for authorship are listed.

### Funding

This research was funded by an International Postdoc grant (2018-06813) by the Swedish Research Council, an UZH Postdoc Grant by the University of Zurich to AGN, an UZH Candoc Grant by the University of Zurich to NB and JK, a Swiss National Science Foundation professorship grant (PP00P3_144816), an European Research Council Starting Grant (SynDegrade; 679881) to MM and an University Research Priority Program 'Adaptive Brain Circuits in Development and Learning' grant by the University of Zurich to NB and MM.

### Acknowledgements

We are grateful to members of the Müller lab, especially Dr Igor Delvendahl, for helpful discussions and critical scientific evaluation of this work. This work is supported by the University Research Priority Program (URPP) 'Adaptive Brain Circuits in Development and Learning (AdaBD)' of the University of Zurich.

### Author's present address

A.G. Nair: Roche Pharma Research and Early Development, Basel, Switzerland. J. Keim: AbbVie AG,Cham, Switzerland.

### Keywords

Drosophila, excitatory synaptic transmission, homeostasis, neuromuscular junction, synaptic plasticity

## Supporting information

Additional supporting information can be found online in the Supporting Information section at the end of the HTML view of the article. Supporting information files available:

**Peer Review History**

