## [Peer Review History · The Journal of Physiology]

Presynaptic Quantal Size Enhancement Counteracts Post-Tetanic Release Depression

Anu G Nair, Nasrin Bollmohr, Levin Schökle, Jennifer Keim, José Maria Mateos, and Martin Mueller
DOI: 10.1113/JP286176

Corresponding author(s): Martin Mueller (martin.mueller@mls.uzh.ch)

The following individual(s) involved in review of this submission have agreed to reveal their identity: Michael Hoppa (Referee #1); Troy Littleton (Referee #2)

Review Timeline:

Submission Date:	25-Apr-2024
Editorial Decision:	21-May-2024
Revision Received:	09-Jul-2024
Accepted:	30-Jul-2024

Senior Editor: Katalin Toth

Reviewing Editor: Samuel Young

Transaction Report:

Dear Dr Mueller,

Re: JP-RP-2024-286176 "Presynaptic Quantal Size Enhancement Counteracts Post-Tetanic Release Depression" by Anu G Nair, Nasrin Bollmohr, Levin Schökle, Jennifer Keim, and Martin Mueller

Thank you for submitting your manuscript to The Journal of Physiology. It has been assessed by a Reviewing Editor and by 2 expert referees and we are pleased to tell you that it is acceptable for publication following satisfactory revision.

REVISION CHECKLIST:

Please upload two versions of your manuscript text: one with all relevant changes highlighted and one clean version with no changes tracked. The manuscript file should include all tables and figure legends, but each figure/graph should be uploaded as separate, high-resolution files. The journal is now integrated with Wiley's Image Checking service. For further details, see: <https://www.wiley.com/en-us/network/publishing/research-publishing/trending-stories/upholding-image-integrity-wileys-image-screening-service>.

- 'Potential Cover Art' for consideration as the issue's cover image
- Appropriate Supporting Information (video, audio or data set: see https://jp.msubmit.net/cgi-bin/main.plex?form_type=display_requirements#supp)

We look forward to receiving your revised submission.

Yours sincerely,

Katalin Toth
Senior Editor
The Journal of Physiology

REQUIRED ITEMS

- Author photo and profile. First or joint first authors are asked to provide a short biography (no more than 100 words for one author or 150 words in total for joint first authors) and a portrait photograph. These should be uploaded and clearly labelled together in a Word document with the revised version of the manuscript. See Information for Authors for further details.

- Please ensure that the Article File you upload is a Word file.

- Your paper contains Supporting Information of a type that we no longer publish, including supplementary tables and figures. Any information essential to an understanding of the paper must be included as part of the main manuscript and figures. The only Supporting Information that we publish are video and audio, 3D structures, program codes and large data files. Your revised paper will be returned to you if it does not adhere to our Supporting Information Guidelines.

- Please include an Abstract Figure file, as well as the Figure Legend text within the main article file. The Abstract Figure is a piece of artwork designed to give readers an immediate understanding of the research and should summarise the main conclusions. If possible, the image should be easily 'readable' from left to right or top to bottom. It should show the physiological relevance of the manuscript so readers can assess the importance and content of its findings. Abstract Figures should not merely recapitulate other figures in the manuscript. Please try to keep the diagram as simple as possible and without superfluous information that may distract from the main conclusion(s). Abstract Figures must be provided by authors no later than the revised manuscript stage and should be uploaded as a separate file during online submission labelled as File Type 'Abstract Figure'. Please also ensure that you include the figure legend in the main article file. All Abstract Figures should be created using BioRender. Authors should use The Journal's premium BioRender account to export high-resolution images. Details on how to use and access the premium account are included as part of this email.

EDITOR COMMENTS

Reviewing Editor:

This manuscript is centered on elucidating the cellular and molecular mechanisms that enable reliable and sustained synaptic transmission. Both reviewers found the findings highly impactful, the data supportive of the conclusions and highly rigorous. Both reviewers agree that more discussion that compares and contrasts their findings in the context of prior reports. Reviewer#1 has specific comments in respect to EM analysis of SV populations and whether the increase in SV size is specific only to those close to the T-bar or are all SVs increased in size. The authors need to address this by analyzing the SVs from their micrographs. Other comments can be addressed by carefully revising and rewriting their manuscript. Supplemental data should be moved to the main text of the manuscript.

REFEREE COMMENTS

Referee #1:

Nair et al., explore a critical question about synaptic transmission, how to reliably respond to high periods of demand (electrical) activity that is quite rapid with a limited resource (synaptic vesicles), whose replenishment is slower and requires reformation and refilling with neuro transmitter. The authors rely on high quality recordings of postsynaptic response from a classic system, *Drosophila* larval NMJ. Nair et al, differentiate evoked responses (EPSPs) from spontaneous release (mEPSPs) to account for response to electrical stimulation compared to the vesicle reformation and refilling. Excellent data suggests that in a dose dependent fashion, the amount of neurotransmitter within a vesicle increases (mEPSPs) rather linearly under increasing loads of stimulation to more or less sustain the magnitude of EPSPs when vesicles depletion is underway. Thus when vesicles numbers are depleted, their magnitude of neurotransmitter content increases to sustain responses of the NMJ. This is an entirely novel presynaptic almost "homeostatic-like" response, which the authors find is dependent on presynaptic machinery, but not postsynaptic glutamate receptors. It is definitely non-canonical that the amount of neurotransmitter per vesicle is variable, but seems well founded by the data. This is really an exciting and interesting finding that would be of broad interest to the scientific community. I would say that the mechanism of vesicle size, rather than filling density or compound vesicle fusion that is offered is less clear, and dims some enthusiasm for the other exciting findings. Details and examples in methods of why only a subset of vesicle diameters are reported, clear highlighted examples of vesicles are needed to fully evaluate data and conclusions. The authors also give very light treatments in the results and discussion about the large increase in mEPSP frequency or report it in some manipulations. Overall, Some points below are suggested to be addressed to improve an exciting paper:

Major points.

1) The methods for identifying vesicles in EM for analysis are not clear, and examples of vesicles measured should be highlighted (color added to inside of vesicles in example EMs as a supplement at a minimum with at least two examples of each condition). Figure 6A (particularly "stim" condition) are too dim to be obvious to a non EM-expert at the least of what constitutes a vesicle. It seems essential to make this clear and to justify why only vesicle diameters near the T-Bar are measured. Other populations should be measured and reported as well at other radius's from the T-Bar. Prior work from Betz and Rizzoli would suggest that proximity to release and vesicle release/reformation are not only at the active zone.

2) Does vesicle size and stimulation frequency correlate at all? Why only compare one condition (why not measure and report 3E?). It is slightly puzzling that mEPSC size distribution and variability increase (Fig 1B), but vesicle diameter variability decreases (Fig. 6B). My understanding is that changes in vesicle volume account for the increased mEPSP magnitude.

3) Why does mEPSC frequency increase for such a long period of time after. Is this also correlated with vesicle diameter. I could not find if the frequency of spont mEPSC was also changed in the genetic and pharmacological manipulations to help understand if mEPSC frequency and magnitude continue to correlate. Is there any chance that multivesicular spontaneous release is occurring with much higher rates of mEPSC fusion?

Minor Points

Line 228 subheading could use some word smithing, "unlikely involves" reads quite awkward. Similar to line 259 of "unlikely plays".

Figures should report p values rather than showing NS or **s in figures for higher rigor and ease of assessing figures.

Referee #2:

In the current manuscript, Nair et al. describe an increase in synaptic vesicle diameter and quantal size following prolonged high frequency stimulation at the *Drosophila* NMJ. The authors find that this increase in quantal size can counteract the

decrease in overall synaptic vesicle density and depression in evoked release during post-tetanic depression. The authors characterize the kinetics and recovery of this adaption, as well as demonstrating it does require modulation of postsynaptic glutamate receptors, but is instead sensitive to manipulation altering SV endocytosis or refilling of SVs with neurotransmitter release. Overall, the data provide an interesting mechanism by which evoked output can be balanced by having increased SV size and neurotransmitter content temporarily balance out a decrease in the number of SVs available to fuse following depletion from a high frequency train. I have only a few minor comments below:

1. It is surprising how quickly the increase mEPSP amplitude is observed after the 80 Hz stimulation. I would assume there are still SVs in the reserve pool that have not been used and would presumably not show increases in size (as they would not have gone through the exo-endo cycle). It seems instead there is preferential reuse of SVs that have fused during the train - this would be the only pool available for endocytic regulation of SV size. Any thoughts on why there would be a rapid reuse of this exocytosed SV pool following the train versus other reserve SV pools that might be available (and that would not have altered size).

2. The authors list the Akbergenova & Bykhovskaia 2009 paper in their reference list, but do not actually cite it in the paper. This one is important to cite and talk a bit about, as it shows a similar phenomena at play with a different type of strong stimulation - high K⁺. The author see a similar increase in SV size and mini amplitude after that stimulation paradigm, and thus highly relevant to the current study.

END OF COMMENTS

Confidential Review

25-Apr-2024

EDITOR COMMENTS

Reviewing Editor:

This manuscript is centered on elucidating the cellular and molecular mechanisms that enable reliable and sustained synaptic transmission. Both reviewers found the findings highly impactful, the data supportive of the conclusions and highly rigorous. Both reviewers agree that more discussion that compares and contrasts their findings in the context of prior reports. Reviewer#1 has specific comments in respect to EM analysis of SV populations and whether the increase in SV size is specific only to those close to the T-bar or are all SVs increased in size. The authors need to address this by analyzing the SVs from their micrographs. Other comments can be addressed by carefully revising and rewriting their manuscript. Supplemental data should be moved to the main text of the manuscript.

We would like to thank the reviewing editor for the constructive feedback. Based on the reviewers' comments, we now analyzed synaptic vesicle diameter outside of T-bars, extended the Discussion to provide more context for our findings, and addressed the remaining points by revising the manuscript. Importantly, the new analysis of synaptic vesicle diameter outside of active zones revealed a post-tetanic increase in synaptic vesicle diameter, thus consolidating our previous quantification at active zones, and suggesting bouton-wide changes in synaptic vesicle size. Detailed responses to each of the reviewers' points are given below.

REFEREE COMMENTS

Referee #1:

Nair et al., explore a critical question about synaptic transmission, how to reliably respond to high periods of demand (electrical) activity that is quite rapid with a limited resource (synaptic vesicles), whose replenishment is slower and requires reformation and refilling with neuro transmitter. The authors rely on high quality recordings of postsynaptic response from a classic system, *Drosophila* larval NMJ. Nair et al, differentiate evoked responses (EPSPs) from spontaneous release (mEPSPs) to account for response to electrical stimulation compared to the vesicle reformation and refilling. Excellent data suggests that in a dose dependent fashion, the amount of neurotransmitter within a vesicle increases (mEPSPs) rather linearly under increasing loads of stimulation to more or less sustain the magnitude of EPSPs when vesicles depletion is underway. Thus when vesicles numbers are depleted, their magnitude of neurotransmitter content increases to sustain responses of the NMJ. This is an entirely novel presynaptic almost "homeostatic-like" response, which the authors find is dependent on presynaptic machinery, but not postsynaptic glutamate receptors. It is definitely non-canonical that the amount of neurotransmitter per vesicle is variable, but seems well founded by the data. This is really an exciting and interesting finding that would be of broad interest to the scientific community. I would say that the mechanism of vesicle size, rather than filling density or compound vesicle fusion that is offered is less clear, and dims some enthusiasm for the other exciting findings. Details and examples in methods of why only a subset of vesicle

diameters are reported, clear highlighted examples of vesicles are needed to fully evaluate data and conclusions. The authors also give very light treatments in the results and discussion about the large increase in mEPSP frequency or report it in some manipulations. Overall, Some points below are suggested to be addressed to improve an exciting paper:

Major points

1) The methods for identifying vesicles in EM for analysis are not clear, and examples of vesicles measured should be highlighted (color added to inside of vesicles in example EMs as a supplement at a minimum with at least two examples of each condition). Figure 6A (particularly "stim" condition) are too dim to be obvious to a non EM-expert at the least of what constitutes a vesicle. It seems essential to make this clear and to justify why only vesicle diameters near the T-Bar are measured. Other populations should be measured and reported as well at other radius's from the T-Bar. Prior work from Betz and Rizzoli would suggest that proximity to release and vesicle release/reformation are not only at the active zone.

We would like to thank the reviewer for the valuable suggestions. We now quantified synaptic vesicle diameter in a region of interest (ROI) outside the T-bar. This analysis yielded an increase in synaptic vesicle diameter at stimulated synapses (Fig. 8D-F), similar to the quantification of the ROI around the T-bar (Fig. 8A-C). Furthermore, we now describe how synaptic vesicle diameters were quantified in greater detail in the methods section (l. 220), highlighted segmented synaptic vesicles in Fig. 4E, and show two examples with better contrast for each condition (Fig. 8A, D). Of note, and as specified in the methods section (l. 228), data were independently analyzed in a blinded fashion by two experimenters. Together, our quantification of different ROIs show that high-frequency stimulation induces an increase in synaptic vesicle diameter at and beyond active zones, suggesting bouton-wide changes in synaptic vesicle size after stimulation.

2) Does vesicle size and stimulation frequency correlate at all? Why only compare one condition (why not measure and report 3E?). It is slightly puzzling that mEPSC size distribution and variability increase (Fig 1B), but vesicle diameter variability decreases (Fig. 6B). My understanding is that changes in vesicle volume account for the increased mEPSP magnitude.

Unfortunately, the timeline allocated to the revisions by the editor was too short to conduct EM-based analysis of synaptic vesicle diameter at different frequencies.

We now compared mEPSP amplitude distributions before and after stimulation, and observed a scaled increase in mEPSP amplitude distributions after tetanic stimulation (Fig. 1) consistent with multiplicative scaling (Turrigiano et al., 1998), suggesting no major changes in mEPSP amplitude distribution after tetanic stimulation.

Fig. 1 Post-tetanic up-scaling of quantal size (A) Sorted mEPSP amplitudes recorded after tetanic stimulation (80 Hz, 30 seconds; Post) vs. mEPSP amplitudes recorded before stimulation (Pre) for a representative NMJ (Post-tetanic mEPSPs were randomly sampled to match mEPSP number before stimulation). A linear regression (dashed line) was fitted to the values. (B) Cumulative frequency distribution of mEPSP amplitudes before stimulation (Pre) and downscaled post-stimulation mEPSP amplitudes (Post). The fitted linear function from A was used to downscale the post-stimulation values. Note that both cumulative histograms overlap, indicating multiplicative synaptic scaling. Similar observations were made in all NMJs.

It is also worth noting that the mEPSP distribution before stimulation is typically more skewed because of a larger fraction of small events that are not detectable, thus appearing narrower (Fig. 1B).

As suggested, we quantified synaptic vesicle diameter in ROIs outside the T-bars (see point 1). We opted against analyzing synaptic vesicle diameter in the ROIs previously used for synaptic vesicle density quantification (previous Fig. 3E) because they partially overlap with the T-bar ROIs. We did not reveal apparent changes in synaptic vesicle diameter distribution in either ROI (Fig. 8B, C, E, F). Hence, tetanic stimulation increased quantal size and vesicle diameter without major changes in the respective distributions.

As discussed in the manuscript (l. 463, l.562), the magnitude of the increase in outer synaptic vesicle diameter upon tetanic stimulation implies that an increase in synaptic vesicle volume is a major contributor to the increase in mEPSP amplitude.

3) Why does mEPSC frequency increase for such a long period of time after. Is this also correlated with vesicle diameter. I could not find if the frequency of spont mEPSC was also changed in the genetic and pharmacological manipulations to help understand if mEPSC frequency and magnitude continue to correlate. Is there any chance that multivesicular spontaneous release is occurring with much higher rates of mEPSC fusion?

The increase in mEPSC frequency after high-frequency stimulation has been studied in detail at the *Drosophila* NMJ in previous investigations (Cho et al., 2015; Jorquera et al., 2012). These previous studies neither investigated synaptic vesicle diameter, nor report mEPSP amplitude changes. Unfortunately, the short revision time did not permit analyzing synaptic vesicle diameter at longer delays after tetanic stimulation,

at which the mEPSP frequency increase is more pronounced than the mEPSP amplitude increase (Cho et al., 2015; Jorquera et al., 2012).

Crucially, pharmacological or genetic dynamin perturbation (Fig. 6), as well as bafilomycin treatment (Fig. 7) largely abolished the post-tetanic increase in mEPSP amplitude, whereas the post-tetanic increase in mEPSP frequency persisted (Fig. 6, 7). Hence, three independent experiments support the conclusion that the post-tetanic increase in miniature amplitude and frequency are separable. Given that multivesicular release is positively correlated with miniature frequency, these results also imply that multivesicular is largely dispensable for the post-tetanic quantal size increase (l. 551).

Minor Points

Line 228 subheading could use some word smithing, "unlikely involves" reads quite awkward. Similar to line 259 of "unlikely plays".

We changed the respective subheading to "No evidence for a role of postsynaptic GluR modulation and Ca^{2+} signaling in post-tetanic quantal size potentiation" (l. 355), and the sentence to "(...) again implying no major involvement for iGluR subunit-specific modulation and post-tetanic quantal size modulation" (l. 375).

Figures should report p values rather than showing NS or *'s in figures for higher rigor and ease of assessing figures.

We now report p values in the figures.

Referee #2:

In the current manuscript, Nair et al. describe an increase in synaptic vesicle diameter and quantal size following prolonged high frequency stimulation at the *Drosophila* NMJ. The authors find that this increase in quantal size can counteract the decrease in overall synaptic vesicle density and depression in evoked release during post-tetanic depression. The authors characterize the kinetics and recovery of this adaption, as well as demonstrating it does require modulation of postsynaptic glutamate receptors, but is instead sensitive to manipulation altering SV endocytosis or refilling of SVs with neurotransmitter release. Overall, the data provide an interesting mechanism by which evoked output can be balanced by having increased SV size and neurotransmitter content temporarily balance out a decrease in the number of SVs available to fuse following depletion from a high frequency train. I have only a few minor comments below:

1. It is surprising how quickly the increase mEPSP amplitude is observed after the 80 Hz stimulation. I would assume there are still SVs in the reserve pool that have not been used and would presumably not show increases in size (as they would not have gone through the exo-endo cycle). It seems instead there is preferential reuse

of SVs that have fused during the train - this would be the only pool available for endocytic regulation of SV size. Any thoughts on why there would be a rapid reuse of this exocytosed SV pool following the train versus other reserve SV pools that might be available (and that would not have altered size).

This is indeed an intriguing possibility. At the *Drosophila* NMJ, ~30% to ~85% of all synaptic vesicles are thought to reside in a reserve pool, which does not undergo synaptic vesicle recycling upon low-frequency stimulation (Delgado et al., 2000; Poskanzer & Davis, 2004). Tetanic stimulation (30 Hz for 180 s) can mobilize and trigger fusion of vesicles from the reserve pool, and newly endocytosed vesicles are incorporated into both pools upon tetanic stimulation at the *Drosophila* NMJ (Kuromi & Kidokoro, 2000). Furthermore, the recruitment of reserve pool vesicles may continue after tetanic stimulation (Kuromi & Kidokoro, 2000). Hence, reserve pool vesicles likely contribute to release after the tetanic stimulation protocol used in our study.

There is evidence that recycling and reserve pool vesicles predominately locate towards the bouton periphery and center at the *Drosophila* NMJ, respectively (Kuromi & Kidokoro, 2000). We therefore analyzed synaptic vesicle diameter in a region of interest outside the active zone/T-bar, which was “randomly” placed within the bouton, including central regions (see Methods). Interestingly, our analysis revealed an increase in synaptic vesicle diameter after tetanic stimulation (Fig. 8E, F), suggesting bouton-wide changes in synaptic vesicle size.

Furthermore, considering the total number of synaptic vesicles per NMJ (~84,000;(Atwood et al., 1993)) and release-ready vesicle resupply rates during sustained activity per NMJ (~1,000-7,500 vesicles/s;(Delgado et al., 2000; Hallermann et al., 2010), our tetanic stimulation protocol is expected to trigger the fusion of a large fraction of the total synaptic vesicle pool.

Our analysis cannot differentiate whether the vesicles released after tetanic stimulation are drawn from a recycling or reserve pool. However, previous work demonstrated that tetanic stimulation recruits reserve pool vesicles (Kuromi & Kidokoro, 2000). Moreover, assuming that the larger synaptic vesicles seen after tetanic stimulation are newly endocytosed, and that reserve pool vesicles locate towards the bouton center, the bouton-wide changes in synaptic vesicle diameter indicate that a significant fraction of vesicles, including reserve pool vesicles, were recycled in response to tetanic stimulation. We now discuss the possibility of a preferential use of newly recycled synaptic vesicles, along with the post-tetanic recruitment of reserve pool vesicles, as well as the bouton-wide changes in synaptic vesicle diameter in detail in the Discussion (l. 499).

2. The authors list the Akbergenova & Bykhovskaia 2009 paper in their reference list, but do not actually cite it in the paper. This one is important to cite and talk a bit about, as it shows a similar phenomena at play with a different type of strong stimulation - high K+. The author see a similar increase in SV size and mini amplitude after that stimulation paradigm, and thus highly relevant to the current study.

We now discuss this relevant paper in the Discussion section (l. 537).

References

- Atwood, H. L., Govind, C. K., & Wu, C. F. (1993). Differential ultrastructure of synaptic terminals on ventral longitudinal abdominal muscles in *Drosophila* larvae. *J Neurobiol*, 24(8), 1008-1024. <https://doi.org/10.1002/neu.480240803>
- Cho, R. W., Buhl, L. K., Volfson, D., Tran, A., Li, F., Akbergenova, Y., & Littleton, J. T. (2015). Phosphorylation of Complexin by PKA Regulates Activity-Dependent Spontaneous Neurotransmitter Release and Structural Synaptic Plasticity. *Neuron*, 88(4), 749-761. <https://doi.org/10.1016/j.neuron.2015.10.011>
- Delgado, R., Maureira, C., Oliva, C., Kidokoro, Y., & Labarca, P. (2000). Size of Vesicle Pools, Rates of Mobilization, and Recycling at Neuromuscular Synapses of a *Drosophila* mutant, *shibire*. *Neuron*, 28(3), 941-953. [https://doi.org/10.1016/S0896-6273\(00\)00165-3](https://doi.org/10.1016/S0896-6273(00)00165-3)
- Hallermann, S., Heckmann, M., & Kittel, R. J. (2010). Mechanisms of short-term plasticity at neuromuscular active zones of *Drosophila*. *Hfsp j*, 4(2), 72-84. <https://doi.org/10.2976/1.3338710>
- Jorquera, R. A., Huntwork-Rodriguez, S., Akbergenova, Y., Cho, R. W., & Littleton, J. T. (2012). Complexin controls spontaneous and evoked neurotransmitter release by regulating the timing and properties of synaptotagmin activity. *J Neurosci*, 32(50), 18234-18245. <https://doi.org/10.1523/jneurosci.3212-12.2012>
- Kuromi, H., & Kidokoro, Y. (2000). Tetanic stimulation recruits vesicles from reserve pool via a cAMP-mediated process in *Drosophila* synapses. *Neuron*, 27(1), 133-143. [https://doi.org/10.1016/s0896-6273\(00\)00015-5](https://doi.org/10.1016/s0896-6273(00)00015-5)
- Poskanzer, K. E., & Davis, G. W. (2004). Mobilization and fusion of a non-recycling pool of synaptic vesicles under conditions of endocytic blockade. *Neuropharmacology*, 47(5), 714-723. <https://doi.org/10.1016/j.neuropharm.2004.07.026>
- Turrigiano, G. G., Leslie, K. R., Desai, N. S., Rutherford, L. C., & Nelson, S. B. (1998). Activity-dependent scaling of quantal amplitude in neocortical neurons. *Nature*, 391(6670), 892-896. <https://doi.org/10.1038/36103>

Dear Dr Mueller,

Re: JP-RP-2024-286176R1 "Presynaptic Quantal Size Enhancement Counteracts Post-Tetanic Release Depression" by Anu G Nair, Nasrin Bollmohr, Levin Schökle, Jennifer Keim, José Maria Mateos, and Martin Mueller

We are pleased to tell you that your paper has been accepted for publication in The Journal of Physiology.

Authors should note that it is too late at this point to offer corrections prior to proofing. Major corrections at proof stage, such as changes to figures, will be referred to the Editors for approval before they can be incorporated. Only minor changes, such as to style and consistency, should be made at proof stage. Changes that need to be made after proof stage will usually require a formal correction notice.

If you would like to receive our 'Research Roundup', a monthly newsletter highlighting the cutting-edge research published in The Physiological Society's family of journals (The Journal of Physiology, Experimental Physiology and Physiological Reports), please click this link, fill in your name and email address and select 'Research Roundup': <https://www.physoc.org/journals-and-media/membernews/>.

Yours sincerely,

Katalin Toth
Senior Editor
The Journal of Physiology

P.S. - You can help your research get the attention it deserves! Check out Wiley's free Promotion Guide for best-practice recommendations for promoting your work at www.wileyauthors.com/eeo/guide. You can learn more about Wiley Editing Services which offers professional video, design, and writing services to create shareable video abstracts, infographics, conference posters, lay summaries, and research news stories for your research at www.wileyauthors.com/eeo/promotion.

IMPORTANT NOTICE ABOUT OPEN ACCESS: To assist authors whose funding agencies mandate public access to published research findings sooner than 12 months after publication, The Journal of Physiology allows authors to pay an Open Access (OA) fee to have their papers made freely available immediately on publication.

You can check if your funder or institution has a Wiley Open Access Account here: <https://authorservices.wiley.com/author-resources/Journal-Authors/licensing-and-open-access/open-access/author-compliance-tool.html>.

EDITOR COMMENTS

Reviewing Editor:

The authors have done an excellent job of responding to the reviewer comments. There are no further concerns.

REFEREE COMMENTS

Referee #1:

The authors have addressed all my concerns with additional experiments and edits. I want to convey my appreciation for their careful work and congratulations on their study.

Referee #2:

The authors have addressed concerns, or reworded aspects of the manuscript, satisfactorily.

1st Confidential Review

09-Jul-2024